# EDiSon: Efficient Design-and-Control Optimization with Reinforcement Learning and Adaptive Design Reuse

## Abstract

Seeking good designs is a central goal of many important domains, such as robotics, integrated circuits (IC), medicine, and materials science. These design problems are expensive, time-consuming, and traditionally performed by human experts. Moreover, the barriers to domain knowledge make it challenging to propose a universal solution that generalizes to different design problems. In this paper, we propose a new method called Efficient Design and Stable Control (EDiSon) for automatic design and control in different design problems. The key ideas of our method are (1) interactive sequential modeling of the design and control process and (2) adaptive exploration and design replay. To decompose the difficulty of learning design and control as a whole, we leverage sequential modeling for both the design process and control process, with a design policy to generate step-by-step design proposals and a control policy to optimize the objective by operating the design. With deep reinforcement learning (RL), the policies learn to find good designs by maximizing a reward signal that evaluates the quality of designs. Furthermore, we propose an adaptive exploration and replay strategy based on a design memory that maintains high-quality designs generated so far. By regulating between constructing a design from scratch or replaying a design from memory to refine it, EDiSon balances the trade-off between exploration and exploitation in the design space and stabilizes the learning of the control policy. In the experiments, we evaluate our method in robotic morphology design and Tetris-based design tasks. Our results show that our method effectively learns to explore high-quality designs and outperforms previous results in terms of design score and efficiency.

## 1 Introduction

Design optimization presents a key challenge across various domains such as robotics (Gupta et al., 2021), integrated circuits (IC) (Mirhoseini et al., 2021), medicine (Coley et al., 2017), and materials science (Ghugare et al., 2023; Govindarajan et al., 2024). Traditionally, design problems are tackled by human experts through iterative manual experimentation, incurring significant costs in both time and resources. Moreover, the required specialized domain knowledge further complicates the design process and increases the need for domain expertise, hindering the generalizability of traditional approaches. Therefore, developing an efficient and general framework for different design problems with little human intervention and specialized domain knowledge is essential.

Recent advancements in reinforcement learning (RL) have made design automation a promising application (Jeong & Jo, 2021; Budak et al., 2022; Dworschak et al., 2022; Govindarajan et al., 2024). RL can rapidly discover and test potential solutions through interacting with design simulators (Sternke & Karpiak, 2023), enabling faster exploration than humans. However, the combinatorial complexity of design space often results in very few valuable designs as well as exponentially many paths to find them (Mouret & Clune, 2015; Colas et al., 2020). In addition to the difficulty of exploring valuable designs in a large and complex space, the challenge is further exacerbated when constructing the design, which is only part of the problem. This occurs when a given design also requires a control policy to achieve its task and evaluate the quality of each design (Gupta et al., 2021). For instance, constructing a robot optimized for locomotion requires both a suitable morphology design and a

Figure 1: The illustration of Efficient Design and Stable Control (EDiSon). The design policy takes steps to generate the design, which is followed by the control policy. Both the design policy and control policy learn from the return signals. Moreover, the design memory selectively stores and reuses the designs to balance the exploration-exploitation with a bandit meta-controller.

control policy that maximizes the robot's locomotion capabilities, inducing a multi-level optimization problem.

In the multi-level optimization problem, we have to address two distinct challenges: (1) Constructing the design as a Markov Decision Process (MDP) with unique transition dynamics and (2) Learning a control policy for that MDP. These problems, while both tractable with reinforcement learning (RL), have different priorities. The first problem focuses on exploring the search space for optimal designs, while the second often suffers from sample inefficiency as each new design may need a newly trained control policy. The interaction between these creates a non-stationary optimization problem requiring additional regularization for better convergence.

To address these challenges, we formulate design optimization as a multi-step MDP and propose a general framework with three key components: the design MDP for design optimization, the control MDP for control optimization, and the design buffer. The design buffer maintains a prioritized queue of high-performing designs, reducing non-stationarity and encouraging exploration-exploitation balance. We employ a bandit-based meta-controller to adjust the exploration probability dynamically, ensuring efficient and adaptive learning. This approach effectively integrates design and control optimization, leveraging past successes while continually seeking new possibilities.

Based on our general framework, we present a practical method for efficient design-and-control automation called Efficient Design and Stable Control (EDiSon), which is illustrated in Figure 1. The design policy iteratively generates designs, maximizing the reward signal from the control policy, thereby guiding optimization toward promising designs. We implement design memory through a buffer that collects high-performing and diverse designs. Our adaptive exploration and replay strategy dynamically balances between creating new designs and refining existing ones, encouraging the emergence of diverse, high-quality designs by effectively leveraging past successes while continually seeking new possibilities. The main contributions of our work are summarized as follows:

- **A General and Efficient RL Framework for Design Optimization:** We introduce an efficient and general framework that integrates design and control optimization into a multi-step MDP. This framework effectively addresses the dual challenges of optimizing both design and control policies, offering a more efficient and comprehensive approach to design automation.

- **Adaptive Exploration-Exploitation Trade-off in Design Optimization:** We introduce a practical method, EDiSon, based on adaptive exploration and design replay. Our method leverages a bandit-based meta-controller to dynamically balance exploration and exploitation, enhancing the efficiency of design-and-control automation. By reusing successful designs from a design buffer, EDiSon ensures continuous improvement and optimal performance.

- **The State-of-the-art Efficiency and Performance across Various Design Tasks:** Through extensive experiments, we demonstrate that EDiSon significantly outperforms existing methods. EDiSon achieves superior results in robotic morphology design and Tetris-based design tasks, showcasing its effectiveness and efficiency.

## 2 RELATED WORK

**Machine Learning for Design** Autonomous design research in robotics has advanced through various approaches that have broadly focused on optimizing morphology and control. Early works proposed evolutionary algorithms to adapt the morphology of rigid body and soft body robots to solve pushing or locomotion tasks (Lipson & Pollack, 2000; Hiller & Lipson, 2012). Subsequent work extended these ideas to learning neural controllers in parallel to the morphology (Bongard & Pfeifer, 2003). Compositional Pattern-producing networks have been shown to be good for discovering new morphologies as they could adapt to the changing number of joints in a robot (Auerbach & Bongard, 2012; Jelisavcic et al., 2019). These works illustrate the progression and integration of morphology and control in autonomous design. In addition to robotics, machine learning (ML) has also been applied to many other design problems, including building design (Sun et al., 2021), as well as materials, molecular and protein design (Govindarajan et al., 2024; Ghugare et al., 2023; Watson et al., 2023) and algorithm design (Co-Reyes et al., 2021). The difference between our problem space and the above prior work is explicitly focusing on problems that include two stages of policy learning: a design stage and a synthesis/policy learning stage.

**Design Optimization with RL** RL has been increasingly applied to design optimization, offering efficient methods for exploring complex design spaces. Sims (1994) pioneered the use of evolutionary algorithms with RL principles to design virtual creatures with adaptable behaviors. Gupta et al. (2021) demonstrated the significant impact of optimized morphologies on learning efficiency for targeted tasks. Yuan et al. (2022) introduced an RL framework integrating transformation and control policies to streamline robot design and operation. Ha (2019) jointly optimized agent embodiment using a population-based REINFORCE algorithm. Schaff et al. (2019) applied RL to update distributions over design parameters. These advancements highlight RL's potential to automate and enhance design optimization. RL has also been applied to many other design problems, including concrete structures (Jeong & Jo, 2021), and electronic placement on microchips (Budak et al., 2022). These prior methods make inroads in using RL for design, but they lack tools to cope with the non-stationarity of the optimization to induce higher-performing solutions. In this work, we include a design buffer for adaptively managing non-stationarity and evaluating over a larger set of tasks than prior methods.

## 3 BACKGROUND

In this section, we briefly review the fundamental background used in our work and describe important aspects of settings with joint design problems and control problems.

**Markov Decision Processes (MDP)** Reinforcement Learning (RL) is typically formulated with the modeling of MDP, where at every time step $t$, the world (including the agent) exists in a state $\mathbf{s}_t \in \mathcal{S}$, where the agent is able to perform actions $\mathbf{a}_t \in \mathcal{A}$. The action to take is determined according to a policy $\pi(\mathbf{a}_t | \mathbf{s}_t)$ which results in a new state $\mathbf{s}_{t+1} \in \mathcal{S}$ and reward $r_t = R(\mathbf{s}_t, \mathbf{a}_t)$ according to the transition probability function $P(\mathbf{s}_{t+1} | \mathbf{s}_t, \mathbf{s}_t)$. The goal of an RL agent is to optimize its policy $\pi$ to maximize the future discounted reward $J(\pi) = \mathbb{E}_{r_0, \dots, r_T} \left[ \sum_{t=0}^{T} \gamma^t r_t \right]$, where $T$ is the max time horizon, and $\gamma$ is the discount factor.

**Design-and-Control Problem** In this paper, we aim to solve design problems, where we need to find a high-quality design and control it to optimize the design objective. Consider such a design problem with a design space $\mathcal{D}$, the purpose of this problem is to find an optimal design $d^\star \in \mathcal{D}$ that maximizes an evaluation function $F : \mathcal{D} \to \mathbb{R}$, i.e., $d^\star = \max_d F(d)$. The evaluation function $F$ is not given a priori and is determined by a control process of design. For a design $d$, a control policy $\pi$ operates with the design that leads to a control score $f_\pi(d)$, while the evaluation function $F(d)$ is defined to be the best control score that can be achieved within a control policy space $\Pi$, i.e.,

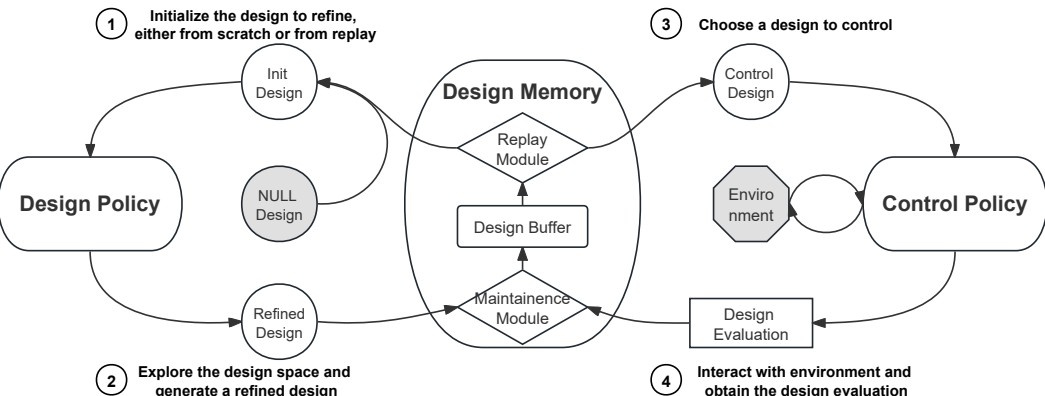

Figure 2: The illustration of our general framework for learning design and control. The framework consists of three components: the design policy, the control policy, and the design memory, which interact with each other as described by the ordered texts.

$F(d) = \max_{\pi \in \Pi} f_\pi(d)$. In real-world applications, one usually aims to find a set of designs that have good evaluation scores and are diverse at the same time.

## 4 A GENERAL FRAMEWORK FOR LEARNING DESIGN AND CONTROL

The design problems we address involve two interconnected challenges: discovering an optimal design (the design problem) and controlling that design to optimize a specific objective (the control problem). This dual challenge is prevalent in scenarios like designing a robotic morphology with a corresponding locomotion policy or creating building blocks for a geometric task. Solving these problems is complex due to the vast combinatorial design space and the intricate landscape of the design objective function. Additionally, control learning must generalize across various designs, further complicating the process. The interplay between design and control exacerbates the difficulty, as design evaluation signals are often noisy and dependent on the ongoing control learning process, while the control problem must handle a non-stationary distribution of designs generated in real time.

To handle these challenges, in this section, we propose a general framework for learning design and control. As illustrated in Figure 2, the framework consists of three components as introduced below.

**Design As A Multi-Step MDP** In this paper, we assume that the Markov assumption holds (see Apendix C Assumption 1) allowing us to formulate the design as a multi-step MDP. The design policy explores the design space and optimizes the design $d \in \mathcal{D}$ regarding the design evaluation signal $F(d)$. We use sequential modeling for the design process, i.e., the design policy starts from an initial base design $d_0$ and constructs it with step-by-step modifications to a final design $d_T$. We define a Design Markov Decision Process (Design MDP) $M = (U, X, P, R, \gamma, \rho, E, D, g)$, where $\mu \in U$ is a state of the design process, $x \in X$ is a design action, $e \in E$ is an optional external information, and $g : D \times X \to D$ describes the deterministic change of design affected by design action:

$$\mu_t \triangleq (d_t, e_t) \qquad \pi^D(x_t \mid \mu_t) \triangleq p(x_t \mid d_t, e_t) \qquad P(\mu_{t+1} \mid \mu_t, x_t) \triangleq \delta_{d_{t+1}} \, p(e_{t+1} | d_t, e_t, x_t)$$

$$\rho(\mu_0) \triangleq p(d_0, e_0) \quad d_{t+1} \triangleq g(d_t, x_t) \qquad R(\mu_t, x_t) \triangleq \begin{cases} F(d_T) & \text{if } t = T \\ 0 & \text{otherwise} \end{cases}$$

$$\tag{1}$$

where $\delta_y$ denotes the Dirac delta distribution with a nonzero density only at $y$.

One key feature of the design-and-control problem is that each design $d$ corresponds to an MDP task to solve, and the design process corresponds to a process of constructing an observation space $\mathcal{O}_d$ and an action space $\mathcal{A}_d$ for the control task. From a finer-grained perspective, the spaces $\mathcal{O}_d, \mathcal{A}_d$ consist of the subspace sets $\{O_i\}, \{A_i\}$, each design action $x_t$ corresponds to adding or removing a tuple of subspaces $(O_i, A_i)$, and the design change function $g$ updates of the subspace sets and generates $\mathcal{O}_d, \mathcal{A}_d$ based on the cartesian product of the subspaces chosen so far. Next, we move on to detail the control task associated with the design $d$ and the observation and action spaces $\mathcal{O}_d, \mathcal{A}_d$ constructed.

**Control As A Multi-Step MDP**   The control policy manipulates a design with the purpose of best performing the control task. Essentially, given a design $d$, this is equivalent to learning the optimal policy in a Control Markov Decision Process (Control MDP) $M_d = (\mathcal{S}_d, \mathcal{A}_d, \mathcal{O}_d, \mathcal{O}, P_d, R_d, \gamma, \rho_d, d)$, where $o \in \mathcal{O}$ is an observation of the environment and $o^d \in \mathcal{O}^d$ is an observation of the design state (e.g., the proprioceptive state of a robot), and $S_d = \mathcal{O} \times \mathcal{O}^d$. Formally, the Control MDP $M_d$ is defined as:

$$s_t \triangleq \left(o_t, o_t^d\right) \qquad P_d\left(s_{t+1} \mid s_t, a_t\right) \triangleq p(o_{t+1}, o_{t+1}^d | o_t, o_t^d, a_t, d)$$
$$\rho_d\left(s_0\right) \triangleq p(o_0, o_0^d) \quad \pi^{\mathrm{C}}\left(a_t \mid s_t, d\right) \triangleq p\left(a_t \mid o_t, o_t^d, d\right) \qquad\qquad R_d\left(s_t, a_t\right) \triangleq r(o_t, o_t^d, a_t, d)$$

Ideally, the control policy maximizes the performance as $\pi^{\mathrm{C}} = \arg\max_\pi J(\pi, M_d)$, which then serves as the design evaluation signal, i.e., $F(d) = J(\pi^{\mathrm{C}}, M_d)$.

**Design Memory**   The design memory maintains a design buffer $\mathcal{B} = \{d_i\}$. The designs generated by the design policy are kept in $\mathcal{B}$ selectively according to their evaluation (i.e., the maintenance module), e.g., with a probability $p(d) \propto F(d)$. Meanwhile, it provides designs for the learning of the design policy and the control policy (i.e., the replay module)

Our framework presents a unified mathematical model for design-and-control problems. Because the co-optimization of an MDP choice and a solution to the chosen MDP is non-stationary, our framework introduced a buffer to store recent high-value designs which also induces control of the non-stationarity of the designs. Specifically, the design memory keeps useful knowledge of diverse sets of best-performing designs to accelerate the learning process. In learning the design policy, the design memory enables the realization of an exploitation-exploration balance in the design space that also helps find good designs efficiently. In the learning of the control policy, the design memory stabilizes the distribution change of design MDPs and reduces the difficulty of learning over multiple designs, thus leading to better design evaluation.

## 5   EFFICIENT DESIGN AND STABLE CONTROL (EDiSON)

In this section, we describe our approach to improving design optimization with RL by actively reusing designs and adaptively balancing the exploration-exploitation trade-off.

### 5.1   JOINT OPTIMIZATION OF DESIGN AND CONTROL USING REINFORCEMENT LEARNING

We leverage reinforcement learning to design the optimization by dividing the task into two distinct stages. The first stage, the design stage, identifies the optimal design for the control task. The second stage, the control stage, utilizes the generated design to complete the task, with RL agents evaluating each design based on reward feedback from the environment.

The optimization objective for the design stage can be formulated as:

$$d^* = \arg\max_{d \in \mathcal{D}} F(d) \tag{2}$$

Where $F$ is the evaluation function for each design $d$. In our method, designs are evaluated during the control stage using a control policy $\pi$, making $F$ dependent on $\pi$: $F = J(\pi, d) = G_{d,\pi} = \mathbb{E}_{\pi,d}\left[\sum_{t=0}^{H} \gamma^t r_t\right]$. Thus, the joint design and control optimization can be formulated as:

$$\begin{aligned}\text{Design Stage:} \quad & d^* = \arg\max_d J\left(\pi, d\right) \\ \text{Control Stage:} \quad & \pi^* = \arg\max_\pi J(\pi, d)\end{aligned} \tag{3}$$

As mentioned in Sec. 4, the agents typically learn two sub-policies, $\pi^D$ and $\pi^C$, to address this joint optimization. The design policy $\pi^D$ generates each design $d_t$ from an initial design $d_0$, and the control policy $\pi^C$ rolls out the control trajectory to evaluate each design.

While methods like Transform2Act (Yuan et al., 2022) have been successful, they often ignore the exploitation and reuse of previously discovered designs, starting from scratch with a less informative $d_0$, leading to inefficiency. In this paper, we propose a new design-and-control paradigm that actively exploits learned designs, enhancing efficiency and performance.

## 5.2 EXPLORATION AND EXPLOITATION IN DESIGN SPACE

In this paper, we propose two general design methods. The first method involves designing from scratch, allowing for greater freedom to explore the entire design space. However, solely exploring the design space without exploiting current designs is often less effective. Therefore, the second method involves designing from good examples $d_{\text{good}}$, enabling the agent to leverage useful and informative designs. This approach closely mirrors human design processes, where we often base our designs on prior work and masterpieces with exemplary performance. In practice, these good examples can be sourced from a design history or provided by humans prior to training.

For fairness, we propose **not to rely on artificially given good examples**. Instead, we let the agents exploit good examples they found throughout the entire learning process. To facilitate this, we implement a design buffer $\mathcal{B}$ to store good designs encountered during training. Whenever the agent needs to design based on an example, it samples a good design $d_{good} \sim \mathcal{P}_{\mathcal{B}}$ from this buffer, wherein $\mathcal{P}_{\mathcal{B}} = \text{softmax}(G_d)$. More implementation details of our design buffer can be found in App. G.

However, solely relying on existing good examples can lead to sub-optimal solutions by failing to explore the design space adequately. Ideally, the agent should first explore the entire design space and, once good designs have been identified, actively exploit these examples to inform further design efforts. To balance exploration and exploitation, we propose a hybrid approach combining two methods: (1) **Exploration:** designing from scratch and (2) **Exploitation:** designing from good examples. During each design stage in training, the agent decides to design from scratch with probability $p$ and to design from good examples with probability $1 - p$. We call this probability $p$ the **design exploration rate** which allows us to control exploration throughout the training process:

$$\begin{cases} \textbf{Exploration:} \text{ Design from Scratch,} & p \\ \textbf{Exploitation:} \text{ Design from Good Examples (Design Reuse),} & 1 - p \end{cases} \tag{4}$$

By adjusting the probability $p$, we can achieve an optimal trade-off between exploration and exploitation in the design optimization problem. Even with a fixed probability $p$, this method outperforms the original Transform2Act which is equivalent to the special case where $p = 1$ and the agent constantly explores the design space from scratch. Our method offers better performance and efficiency, demonstrating the benefits of integrating both exploration and exploitation in the design process.

## 5.3 ADAPTIVE EXPLORATION IN DESIGN OPTIMIZATION

A fixed probability $p$ helps balance exploration and exploitation but fails to let agents adaptively choose the best design method during different learning stages. Early in training, agents should explore widely using a higher $p$, while later stages should exploit good designs with a lower $p$.

To address this, we propose a meta-controller that dynamically adjusts the design exploration rate $p$, balancing exploration and exploitation. We use a multi-armed bandit (MAB) approach, where each bandit has two arms: arm = 0 for design from scratch and arm = 1 for design from good examples. At the start of each trajectory, the actor samples an arm $k \in K = \{0, 1\}$ using the probability distribution $\mathcal{P}_K = \frac{e^{\text{Score}_k}}{\sum_j e^{\text{Score}_j}}$. The design exploration rate $p$ is given by $p = \mathcal{P}_{arm=0}$.

We use the Upper-Confidence Bound (UCB) score to manage the trade-off:

$$\text{Score}_k = V_k + c \cdot \sqrt{\frac{\log\left(1 + \sum_{j \neq k}^{K} N_j\right)}{1 + N_k}} \tag{5}$$

where $N_k$ is the number of visits to arm $k$, $V_k$ is the expected value of the returns, and the UCB term (i.e., the second term) ensures the agent doesn't repeatedly select the same arm, avoiding quick convergence to suboptimal solutions.

After sampling an arm, the agent decides whether to reuse a base design from the buffer $\mathcal{B}$ or design from scratch. The design policy $\pi^{\text{D}}$ and control policy $\pi^{\text{C}}$ are applied to obtain a trajectory $\tau_i$ and the return $G_i$, which updates the reward model $V_k$ for the selected arm. To handle non-stationarity, we ensemble several MABs with different hyperparameters, allowing the agent to adapt to changing environments and maintain robust performance. More details are in the App. F.

## 5.4 EFFICIENT DESIGN AND STABLE CONTROL (EDISON) ALGORITHM

We summarize the complete process of our method in Algorithm 1, which illustrates the core steps of the Efficient Design and Stable Control (EDiSon) framework. The algorithm iterates over multiple design and control steps, dynamically adjusting between exploration and exploitation, and refining the policies over time to converge on an optimal design and control policy.

---

**Algorithm 1** EDiSon

---

**Require:** number of training iterations $N$, simple initial design $d_{null}$, initial design $d_0$, design buffer $\mathcal{B}$, bandit MAB, design policy $\pi^D$, control policy $\pi^C$, length of design stage $T$
1: Initialize design policy $\pi^D$ and control policy $\pi^C$
2: Initialize design buffer $\mathcal{B} \leftarrow (design = d_{null}, value = 0)$
3: Initialize training data replay buffer $\mathcal{M} \leftarrow \emptyset$
4: **for** iteration $i = 1$ to $N$ **do**
5:    **while** not reaching batch size **do**
6:       **for** jth trajectory $\tau_j$ **do**
7:          // Design Stage
8:          Sample arm $k_j$ from the bandit MAB;
9:          **if** $k_j = 0$ **then**
10:          $d_0 \leftarrow d_{null}$                        ▷ Design from scratch;
11:          **else**
12:          $d_0 \leftarrow$ Sample from Buffer($\mathcal{B}$)            ▷ Design Reuse
13:          **end if**
14:          **for** iteration $t = 1$ to $T$ **do**
15:             Sample design actions $a_t^d$ using $\pi^D$
16:             Update design $d_t$ with sampled actions $a_t^d$
17:          **end for**
18:          // Control Stage
19:          Use $\pi^C$ to rollout control trajectory with design $d_T$, obtain trajectory return $G_j$
20:          Store trajectory $j$ in data replay buffer $\mathcal{M} \leftarrow \tau_j$
21:          Update design buffer $\mathcal{B} \leftarrow (design = d_T, value = G_j)$
22:          Update bandit with $(k_j, G_j)$
23:       **end for**
24:    **end while**
25:    Update $\pi^C$ and $\pi^D$ using PPO with samples from $\mathcal{M}$
26: **end for**
27: **return** Optimal design $d^*$, control policy $\pi^C$, design policy $\pi^D$

---

## 6 EXPERIMENTAL RESULTS

Our experiments are designed to evaluate the effectiveness of our methods across various design optimization tasks, from robotic morphology design to microfabrication-inspired problems. Specifically, we explore Tetris-like design challenges, where a set of designed blocks is manipulated to achieve either a Tetris or target deposition pattern. We propose to address the following questions:

1. How does EDiSon perform compared to prior work in various design tasks (See Figure 3)? Can our methods find better designs (See Figure 5)?

2. How much does adaptively balancing the exploration and exploitation in design optimization assist in finding higher-value solutions (See Figure 6)? Why not just use a fixed design exploration rate $p$ (See Figure 6)?

3. How much do core components of our framework, such as design reuse and adaptive exploration-exploitation trade-off, contribute to the results (See Figure 7)?

### 6.1 EXPERIMENTAL SETUP

We conduct experiments across several design-based tasks, including robotic morphology design and Tetris-based design problems. To ensure a fair comparison, we follow the same settings and network

structure for the robotic morphology design tasks as Transform2Act (Yuan et al., 2022) and adopt a 3-layer MLP for all policies and critics in the Tetris-related task. We use PPO (Schulman et al., 2017) to learn both our design policy, control policy, and critics. We utilize a separate evaluation process to continuously record scores, measuring the undiscounted episodic returns averaged over five seeds. To provide comprehensive insights, we present full learning curves for each task, addressing any issues associated with aggregated metrics. In addition to the average score, we highlight the best designs discovered by our agent during the learning process, showcasing our method's superiority in design exploration. More implementation details can be found in App. I.

**Environments.** We evaluate our algorithm on the following tasks: **(1)** Swimmer: A 2D agent operating in water with 0.1 viscosity, confined to the xy-plane, aiming to maximize forward speed along the x-axis. **(2)** 2D Locomotion: A 2D agent in the xz-plane that moves forward as quickly as possible, with rewards based on forward velocity. **(3)** 3D Locomotion: A 3D agent navigating along the x-axis, striving for maximum forward speed, rewarded based on velocity. **(4)** Gap Crosser: A 2D agent navigating across periodic gaps on the xz-plane, with rewards linked to forward speed. Additionally, we provide supplementary results for other design tasks, such as Tetris rewarded by playtime (i.e., design blocks to play Tetris longer) and Microfabrication Deposition rewarded by matching rate (i.e., design blocks to etch the deposition layers and match target pattern better) to further demonstrate our method's capabilities beyond robot design tasks (see App. L). More details about these tasks can be found in App. D.

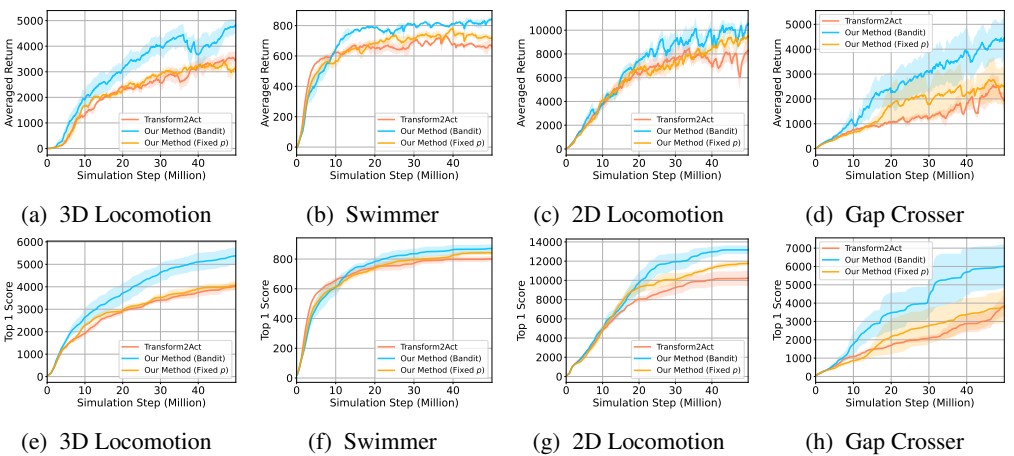

(a) 3D Locomotion    (b) Swimmer    (c) 2D Locomotion    (d) Gap Crosser

(e) 3D Locomotion    (f) Swimmer    (g) 2D Locomotion    (h) Gap Crosser

Figure 3: **Baseline Comparison in Robotic Morphology Design Tasks.** The upper panel (i.e., *a-d*) is the comparison in terms of average return, and the lower panel (i.e., *e-h*) is for the score of best design discovered (Top 1 Score). For each robot task, we plot the mean and standard deviation of total rewards against the number of simulation steps for all methods. Each curve shows a smoothed moving average over 5 points.

## 6.2 SUMMARY OF RESULTS

Our experimental results in Figure 3 demonstrate the superiority of our proposed methods over the baseline, Transform2Act. The Bandit approach consistently achieves higher returns across all tasks, illustrating its effectiveness in dynamically balancing exploration and exploitation. This adaptability is crucial for optimizing performance in varied and complex environments. While the fixed design exploration parameter $p$ also shows improvements, it remains inferior to the Bandit method, underscoring the importance of an adaptive balance in design optimization. The success of our methods can be attributed to several key factors: (1) **Design Reuse**: By leveraging effective designs discovered during the training process, our methods avoid the inefficiencies of always starting from scratch. Reusing successful designs enhances learning efficiency and accelerates performance improvements. (2) **Adaptive Trade-off**: The Bandit method enables the agent to dynamically adjust its exploration-exploitation balance during design optimization, leading to more efficient learning and higher performance. This adaptability ensures that the agent explores new designs early in training and exploits successful designs as they are discovered.

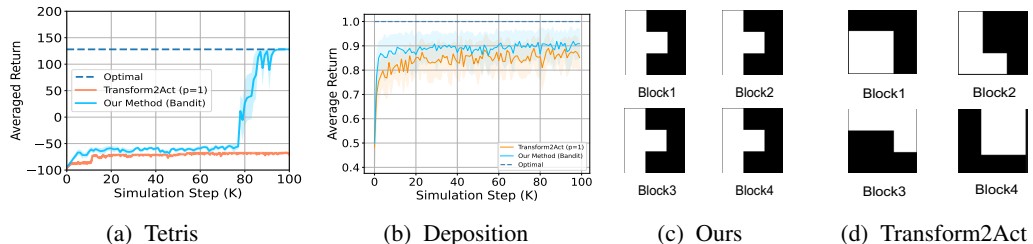

(a) Tetris      (b) Deposition      (c) Ours      (d) Transform2Act

Figure 4: **Baseline Comparison and Best Design Discovered in Tetris-Based Tasks.** (a) and (b) show the learning curve in Tetris-like Tasks. (c) and (d) show the best design in Tetris Tasks, where agents have to find 4 blocks, each represented as a 3 × 3 grid with 4 squares filled (the white one).

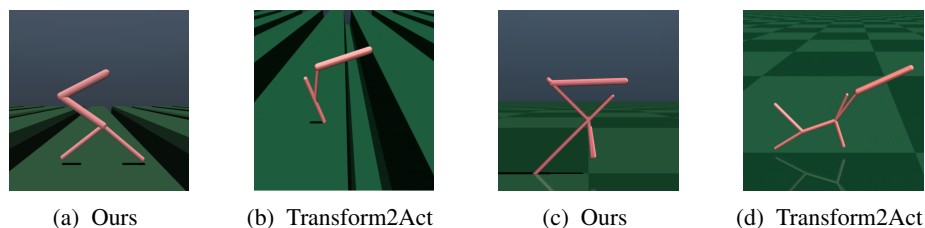

(a) Ours      (b) Transform2Act      (c) Ours      (d) Transform2Act

Figure 5: **Best Design Discovered in Robotic Morphology Design Tasks.** (a) and (b) show the best designs found in the Gap Crosser task by our method (reward: 11572) and Transform2Act (reward: 4579). (c) and (d) illustrate the best designs found in the 2D Locomotion task by our method (reward: 15459) and Transform2Act (reward: 11416). More discovered designs can be found in App. E.

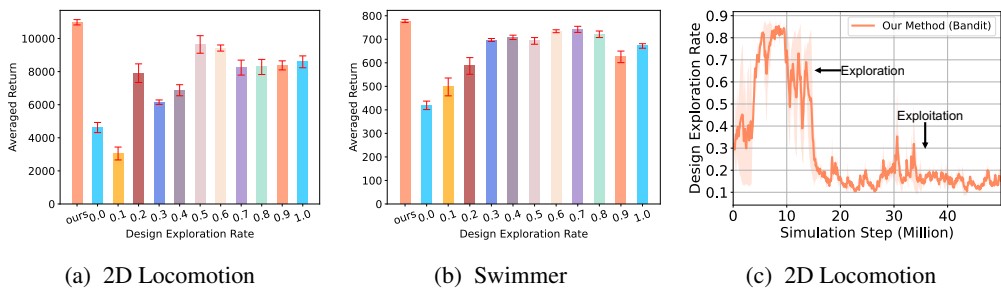

(a) 2D Locomotion      (b) Swimmer      (c) 2D Locomotion

Figure 6: **Case Study Results.** For each robot task, we plot the mean and standard deviation of total rewards against the number of simulation steps for all methods.

Similar results are observed in Tetris-related design tasks in Figure 4, where our method stabilizes learning curves, as detailed in Appendix L. Additionally, in the Microfabrication Deposition tasks shown in Figure 4, our method achieves better final performance than the Transform2Act method, demonstrating our effectiveness and adaptability across a range of tasks.

Further investigation into the best designs found by our methods can also help us to understand the results, which has been illustrated in Figure 5. In the Gap Crosser Task, our bipedal design (Figure 5a) offers enhanced stability and efficiency with its upright posture and elongated limbs, enabling better gap navigation than the sprawled configuration of Transform2Act's design (Figure 5b). For the 2D Locomotion Task, our design (Figure 5c) optimizes limb placement by reducing an unnecessary joint on the tail foot and adding one to the forelimb, resulting in improved speed and agility. Conversely, Transform2Act's design (Figure 5d) retains an additional hind limb, which seems less efficient. Overall, our designs are more structurally optimized for their respective tasks. For the Tetris task, our method outperforms Transform2Act by discovering four identical symmetric block structures. Our blocks simplify the learning of the control policy, facilitate continuous gameplay, and enable efficient line clearing. A more detailed analysis can be found in App. E.3.

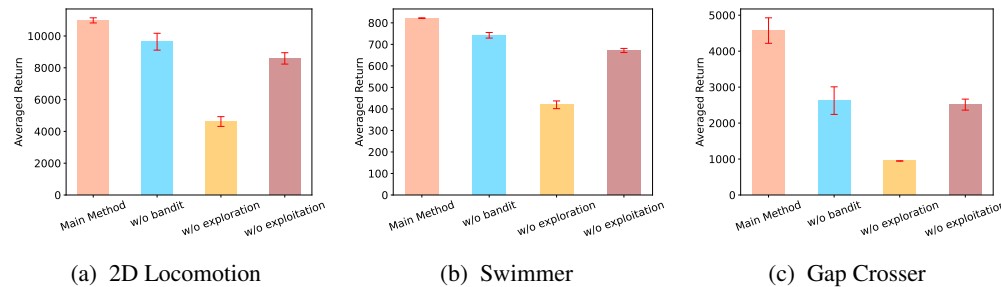

(a) 2D Locomotion     (b) Swimmer     (c) Gap Crosser

Figure 7: **Ablation Study Results.** The mean and standard deviation of each method over 5 random seeds are plotted. Note that Main Method means EDiSon (ours).

### 6.3 CASE STUDY: EXPLORATION-EXPLOITATION TRADE-OFF

We divided the design exploration rate $p$ into ten equal intervals from 0 to 1, creating methods with different exploration preferences. These methods ranged from extreme exploitation ($p = 0$) to extreme exploration ($p = 1$, corresponding to Transform2Act). The results in Figures 6a and 6b show that different tasks have distinct optimal design exploration rates. This variability underscores that achieving a balance between exploration and exploitation is non-trivial and crucial for success.

Additionally, we analyzed the design exploration rate control curve of our Bandit-based method (Figure 6c). The results demonstrate that our Bandit-based meta-controller effectively adjusts the exploration-exploitation trade-off dynamically. Our method promotes extensive exploration during early training stages, which helps discover diverse and potentially optimal designs. As training progresses, the meta-controller gradually shifts towards exploitation, utilizing the accumulated design knowledge to optimize performance. This adaptability ensures that the agent efficiently explores the design space and exploits successful designs, leading to superior performance across tasks.

### 6.4 ABLATION STUDIES

In our ablation studies, we examine two critical components: the adaptive exploration-exploitation trade-off and design reuse via the design buffer. We evaluate several variants to highlight their impact: (1) Ours w/o Bandit: Removes the adaptive mechanism. (2) Ours w/o Exploitation: Eliminates the design buffer, requiring designs from scratch. (3) Ours w/o Exploration: Sets $p$ to 0, disabling exploration. (4) Our Main Method: Incorporates both components.

Figure 7 shows that both design reuse and adaptive exploration-exploitation are crucial. The design buffer leverages successful designs, and the adaptive mechanism balances exploration and exploitation, enhancing performance. Neither extreme exploration nor exploitation is optimal; a balanced approach, as in our main method, yields the best results, highlighting the importance of balancing these factors in design optimization tasks.

## 7 CONCLUSION AND DISCUSSION

In this paper, we presented a novel reinforcement learning framework for design optimization, demonstrating its effectiveness across tasks ranging from robotic morphology design to Tetris-based design challenges. Our Bandit-based meta-controller dynamically balances exploration and exploitation, significantly outperforming existing methods like Transform2Act. Extensive experiments highlight the importance of adaptive strategies and design reuse, revealing the limitations of a fixed exploration rate for complex design problems. Our key contributions include an adaptive exploration-exploitation mechanism, design reuse through a design buffer, and robust evaluation via comprehensive case studies. These advancements enhance performance and efficiency, paving the way for future research in design automation and impacting various domains, from robotics to material science. However, our work has limitations. The computational complexity of our meta-controller might limit its application in resource-constrained environments. Additionally, the quality and diversity of the design buffer are crucial; a lack of initial diversity could compromise performance. Future work should address these limitations to further refine and extend the applicability of our approach.

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

## A  ADVANTAGE OF EFFICIENT DESIGN AND STABLE CONTROL (EDiSon) OVER TRANSFORM2ACT

There are three main advantages of our method (EDiSon) over Transform2Act (Yuan et al., 2022):

1. **Adaptive Exploration-Exploitation Balance:** Transform2Act uses a fixed exploration rate, which is suboptimal for complex design problems. Our method introduces a Bandit-based meta-controller that dynamically adjusts the exploration-exploitation trade-off. This adaptive strategy allows for extensive exploration in the early stages and efficient exploitation of successful designs in later stages, leading to superior performance across various tasks, as demonstrated in our experimental results (see Figures 3 and 14).

2. **Design Reuse with a Design Buffer:** Unlike Transform2Act, which always starts from scratch, our method leverages a design buffer to store and reuse successful designs. This approach enhances learning efficiency by building upon previously discovered high-quality designs. The use of a design buffer facilitates better generalization and reduces the time required to achieve optimal performance, as evidenced by our experimental results.

3. **Increased Exploration Capability:** Our method allows for more extensive exploration of design possibilities in each episode. By dynamically adjusting the exploration rate and leveraging the design buffer, our approach can try a wider variety of designs within a shorter period. This increased exploration capability enables our method to discover innovative and high-performing designs more effectively than Transform2Act, leading to enhanced overall performance and efficiency in design optimization tasks (see Figure 14).

# B  DISCUSSION AND LIMITATIONS

While our method demonstrates significant improvements in design and control automation, it is not without limitations. One notable limitation is the computational complexity associated with our bandit-based meta-controller. The dynamic balancing of exploration and exploitation requires substantial computational resources, which may not be readily available in all settings. This could limit the scalability and applicability of our approach to resource-constrained environments.

Another limitation lies in the assumptions made by our method. Our approach assumes that the design and control tasks can be adequately represented within the framework of a multi-armed bandit problem. This assumption may not hold in all scenarios, particularly in highly complex and dynamic environments where the relationships between design choices and performance outcomes are non-linear and unpredictable. As a result, the effectiveness of our method may vary across different tasks and domains.

Additionally, our method relies heavily on the quality and diversity of the design buffer. If the initial set of designs is not sufficiently diverse or representative of the optimal design space, the performance of our method could be adversely affected. Ensuring the robustness of the design buffer through careful selection and continuous updating is essential to maintain the efficacy of our approach.

In general, our experimental evaluation is limited to specific tasks and environments, and while our results are promising, further validation is needed across a broader range of applications. Future work should explore the generalizability of our method to other design and control problems, as well as investigate potential enhancements to address the identified limitations. By doing so, we aim to refine our approach and extend its applicability to a wider array of real-world challenges.

## C  DESIGN OPTIMIZATION AS MULTI-STEP MDP

In this section, we describe the Markov Decision Processes (MDP) used to formalize the design and control stages of our framework. Using the robotic morphology design with the *Transform2Act* approach (Yuan et al., 2022) as an example, we demonstrate how our formalizations can be applied to analyze an existing design problem and an RL method for design optimization.

**Assumption 1** (Markov Assumption of Design Optimization). *We assume that the design optimization problems we study are all Markovian, meaning that the future state depends only on the current state and action and not on the sequence of events that preceded it. Formally, this is expressed as:*

$$P\left(s_{t+1} \mid s_t, a_t\right) = P\left(s_{t+1} \mid s_t, a_t, s_{t-1}, a_{t-1}, \ldots, s_0, a_0\right). \tag{6}$$

### C.1  DESIGN AS MARKOV DECISION PROCESS

We model the design optimization process as a multi-step Markov Decision Process (MDP), enabling a structured approach to the design stage within our reinforcement learning framework. The elements of this MDP are defined as follows:

**State** $s_t$ :  The state at time $t$ is represented by $s_t \triangleq (d_t, o_t)$, where $d_t$ denotes the design at the time step $t$, and $o_t$ represents the state information of the task/environment. It's worth noting that, when the design is fully represented by $d_t$ and no more other observation can be obtained from the environment, $o_t$ can be ignored.

**Action** $a_t$ :  The action at time $t$ is given by $a_t \triangleq x_{t+1}$, where $x_{t+1}$ indicates the next/target design parameters. This allows the agent to modify the design iteratively.

**Policy** $\pi\left(a_t \mid s_t\right)$ :  The design policy maps the state to actions, which can be defined as $\pi\left(a_t \mid s_t\right) \triangleq p_\theta\left(x_{t+1} \mid d_t, o_t\right)$, where $p_\theta$ is the probability distribution over the actions conditioned on the current state and design.

**State Transition** $P\left(s_{t+1} \mid s_t, a_t\right)$ :  The transition probability is given by $P\left(s_{t+1} \mid s_t, a_t\right) \triangleq \left(\delta_{d_t}, \delta_{o_t}, \delta_{x_{t+1}}\right)$, where $\delta$ denotes the Dirac delta function, ensuring deterministic transitions between states based on the selected actions.

**Initial State Distribution** $\rho_0\left(s_0\right)$ :  The initial state distribution is defined as $\rho_0\left(s_0\right) \triangleq (p(d_0), p(o_0))$, where $p(d_0)$ is the initial design distribution (**which can be controlled by the design exploration rate** $p$), and $p(o_0)$ represents the initial distribution of the initial state information from the environment/task.

**Reward Function** $R\left(s_t, a_t\right)$ :  The reward function is defined as:

$$R\left(s_t, a_t\right) \triangleq \begin{cases} r\left(d_T\right) & \text{if } t = T \\ 0 & \text{otherwise} \end{cases} \tag{7}$$

Here, $r\left(d_T\right)$ evaluates the quality of the final design $d_T$. The design reward signal is sparse, because the agent does not know how well it performs until the control stage has been conducted.

**Definition C.1** (Design Optimization as a MDP). *Based on the above, we formulate the design optimization procedure to the following:*

$$s_t \triangleq (d_t, o_t) \quad \pi\left(a_t \mid s_t\right) \triangleq p_\theta\left(x_{t+1} \mid o_t, d_t\right) \quad P\left(s_{t+1} \mid s_t, a_t\right) \triangleq \left(\delta_{d_t}, \delta_{o_t}, \delta_{x_{t+1}}\right)$$
$$a_t \triangleq x_{t+1} \qquad \rho_0\left(s_0\right) \triangleq (p(d_0), p(o_0)) \quad R\left(s_t, a_t\right) \triangleq \begin{cases} r\left(d_T\right) & \text{if } t = T \\ 0 & \text{otherwise} \end{cases} \tag{8}$$

*in which $\delta_y$ is the Dirac delta distribution with nonzero density only at $y$. In this MDP, trajectories consist of $T$ time steps, leading to a termination state/design. The cumulative reward of each trajectory equals $r\left(d_T\right)$, making the maximization of the design reward $\mathcal{J}_{design}\left(\theta\right)$ equivalent to optimizing the reinforcement learning objective $\mathcal{J}_{\mathrm{RL}}(\pi)$ in this MDP context.*

In the following, we provide the our multi-step MDP framework for design optimization to interpret the design stage of Transform2Act. It is important to note that, for fairness, our main method in robot-related tasks maintains similar Skeleton Transform and Attribute Transform stages as Transform2Act, **except for incorporating design reuse with a design buffer and a bandit-based meta-controller**. In other words, our approach, which includes design reuse and the bandit-based meta-controller, **can be applied to any existing design optimization method using RL**.

**Robotic Morphology Design in Transform2Act**   Transform2Act divides the design stage into two parts, the *Skeleton Transform:* construct the joint structure graph of the robot, and the *Attribute Transform*: fine-tune relevant parameters such as the length of each joint structure.

In the Skeleton Transform stage, the agent follows the policy $\pi_\theta^S \left( a_t^S \mid d_t, \Phi_t \right)$ to modify the skeletal structure. Here, $d_t = (V_t, E_t, A_t)$ includes the skeletal graph $(V_t, E_t)$ and joint attributes $A_t$. $\Phi_t$ is a flag used to indicate the current stage (e.g., Skeleton Transform, Attribute Transform, Control) and can be regarded as part of the environment state $o_t$. The skeleton transform action $a_t^S = \left\{ a_{u,t}^S \right\}_{u \in V_t}$ changes the skeletal graph by adding or deleting joints.

The agent follows the skeleton transform sub-policy $\pi_\theta^S$ for $N_s$ timesteps, resulting in an updated design $d_{t+1} = (V_{t+1}, E_{t+1}, A_{t+1})$, and the policy $\pi_\theta^S$ can be write as:

$$\pi_\theta^S \left( a_{u,t}^S \mid d_t, \Phi_t \right) = \prod_{u \in V_t} \pi_\theta^S \left( a_{u,t}^S \mid d_t, \Phi_t \right) \tag{9}$$

Since Transform2Act always design from scratch, the initial design distribution $p(d_0)$ deterministic distribution:

$$d_0 \sim p(d_0) \triangleq d_{Null} \tag{10}$$

And the total steps of attribute transform stage is $T_S$.

In the Attribute Transform stage, the agent modifies joint attributes using the policy $\pi_\theta^A \left( a_t^A \mid d_t, \Phi_t \right)$. The attribute transform action $a_t^A = \left\{ a_{u,t}^A \right\}_{u \in V_t}$ adjusts continuous attributes like bone length, size, and motor strength. The attribute transform sub-policy $\pi_\theta^A \left( a_{u,t}^A \mid d_t, \Phi_t \right)$ adopts the same GNN-based network as the skeleton transform sub-policy $\pi_\theta^S$. The policy distribution for the attribute transform action is defined as:

$$\pi_\theta^A \left( a_{u,t}^A \mid d_t, \Phi_t \right) = \mathcal{N} \left( a_{u,t}^A; \mu_{u,t}^A, \Sigma^A \right) \tag{11}$$

Here, $\mu_{u,t}^A$ and $\Sigma^A$ are shared by all joints. The new design becomes $d_{t+1} = (V_t, E_t, A_{t+1})$ where the skeleton $(V_t, E_t)$ remains unchanged. And the total steps of attribute transform stage is $T_A$.

The reward signal is sparse for each design step, where only the final reward $r_T$ the final design $d_T$ to achieve the robot control task with control policy $\pi_c$ is given as the learning signal.

## C.2   Control As Markov Decision Process

In this part, we describe the control optimization process as a multi-step Markov Decision Process (MDP), providing a structured approach to the control stage within our reinforcement learning framework. The design evluation is achieved in the control stage, where the agents will interact with the task using the final design and control policy $\pi_c$. The elements of this MDP are defined as follows:

**State** $s_t$ :   The state at time $t$ is represented by $s_t \triangleq (d_T, o_t)$, where $d_T$ denotes the final design of design stage, $o_t$ is the current environment observation.

**Action** $a_t$ :   The action at time $t$ is given by $a_t \triangleq c_{t+1}$, where $c_{t+1}$ indicates the next control parameters. This allows the agent to iteratively modify the control strategy.

**Policy** $\pi\left(a_t \mid s_t\right)$ **:** The policy maps the state to actions, defined as $\pi\left(a_t \mid s_t\right) \triangleq p_\theta\left(c_{t+1} \mid d_T, o_t, c_t\right)$, where $p_\theta$ is the probability distribution over the actions conditioned on the current state and design.

**State Transition** $P\left(s_{t+1} \mid s_t, a_t\right)$ **:** The transition probability is given by $P\left(s_{t+1} \mid s_t, a_t\right) = p(o_{t+1} \mid o_t, d_T, c_{t+1})$ is given by the environment (task-wise).

**Initial State Distribution** $\rho_0\left(s_0\right)$ **:** The initial state distribution is defined as $\rho_0\left(s_0\right) \triangleq (d_T, p(o_0), p(c_0))$, where $d_T$ is the final design, $p(o_0)$ is the initial observation from the environment (task-wise), and $p(c_0)$ represents the initial control parameters.

**Reward Function** $R\left(s_t, a_t\right)$ **:** The reward function is defined as:

$$R\left(s_t, a_t\right) \triangleq r(c_{t+1}, d_T, o_t) \tag{12}$$

Here, $r(c_{t+1}, d_T, o_t)$ is given by the environemnt, just the well-known environment reward in also conditioned on our final design $d_T$.

**Definition C.2** (Control Optimization as a MDP). *Based on the above, we formulate the design optimization procedure to the following:*

$$
\begin{aligned}
s_t &\triangleq (d_T, o_t, c_t) & \pi\left(a_t \mid s_t\right) &\triangleq p_\theta\left(c_{t+1} \mid c_t, d_T\right) & P\left(s_{t+1} \mid s_t, a_t\right) &= p(o_{t+1} \mid o_t, d_T, c_{t+1}) \\
a_t &\triangleq c_{t+1} & \rho_0\left(s_0\right) &\triangleq (d_T, p(o_0), p(c_0)) & R\left(s_t, a_t\right) &\triangleq r(c_{t+1}, d_T, o_t)
\end{aligned}
\tag{13}
$$

*In this MDP, trajectories consist of $T_c$ time steps, leading to a termination control state. The cumulative reward of each trajectory equals $R(\tau) = \sum_{t=0}^{T_c}[r_t]$, making the maximization of the control reward $\mathcal{J}_{control}\left(\theta\right)$ equivalent to optimizing the reinforcement learning objective $\mathcal{J}_{\mathrm{RL}}(\pi)$ in this MDP context.*

**Robot Control of Transform2Act** After the agent performs $T_S$ skeleton transform and $T_A$ attribute transform actions, it enters the control stage where the agent assumes the transformed design and interacts with the environment. A GNN-based execution policy $\pi_\theta^e\left(a_t^e \mid s_t^e, d_t, \Phi_t\right)$ is used in this stage to output motor control actions $a_t^e$ for each joint.

Since the agent now interacts with the environment, the policy $\pi_\theta^e$ is conditioned on the environment state $s_t^e$ as well as the transformed design $d_t$, which affects the dynamics of the environment. The control actions are continuous. The execution policy distribution is defined as:

$$\pi_\theta^e\left(a_{u,t}^e \mid s_t^e, d_t, \Phi_t\right) = \mathcal{N}\left(a_{u,t}^e; \mu_{u,t}^e, \Sigma^e\right) \tag{14}$$

where the environment state $s_t^e = \left\{s_{u,t}^e \mid u \in V_t\right\}$ includes the state of each node $u$ (e.g., joint angle and velocity). The GNN uses the environment state $s_t^e$ and joint attributes $A_t$ as input node features to output the mean $\mu_{u,t}^e$ of each joint's Gaussian action distribution. $\Sigma^e$ is a state-independent learnable diagonal covariance matrix shared by all joints. The agent applies the motor control actions $a_t^e$ to all joints and the environment transitions the agent to the next environment state $s_{t+1}^e$ according to the environment's transition dynamics $\mathcal{T}^e\left(s_{t+1}^e \mid s_t^e, a_t^e\right)$. The design $d_t = d_{T_S+T_A}$ remains unchanged throughout the control stage.

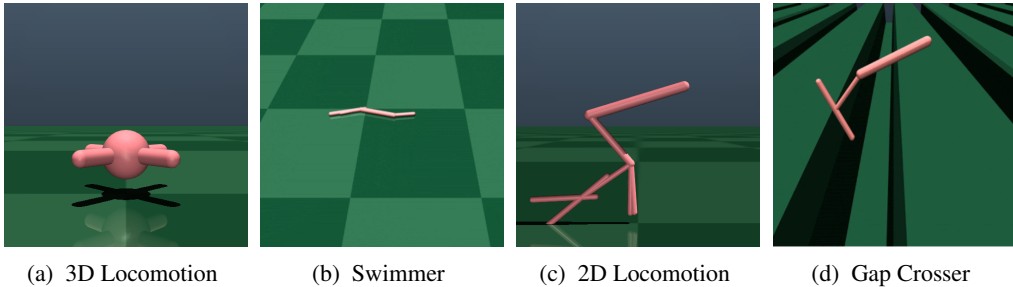

(a) 3D Locomotion   (b) Swimmer   (c) 2D Locomotion   (d) Gap Crosser

Figure 8: A random agent in each of four different taks.

## D  ENVIRONMENT DETAILS

### D.1  ROBOT-RELATED TASK

In this part, we provide a comprehensive overview of the four robot-related environments used in our experiments.

#### D.1.1  2D LOCOMOTION

The agent in this environment operates within an $xz$-plane with flat ground at $z = 0$. Each joint of the agent can have up to three child joints. For the root joint, additional features such as height and 2D world velocity are included in the state representation. The reward function is defined as:

$$r_t = \frac{|x_{t+1} - x_t|}{\Delta t} + 1, \tag{15}$$

where $x_t$ represents the $x$-position of the agent and $\Delta t = 0.008$ is the time step. An alive bonus of 1 is also incorporated into the reward. The episode terminates when the root height drops below 0.7 .

#### D.1.2  3D LOCOMOTION

In this environment, the agent operates in a 3D space with flat ground at $z = 0$. Similar to the 2D Locomotion, each joint can have up to three child joints, with the root joint including height and 3D world velocity in its state representation. The reward function is given by:

$$r_t = \frac{|x_{t+1} - x_t|}{\Delta t} - \alpha \cdot \frac{1}{N} \sum_{i=1}^{N} \|a_{i,t}\|^2 \tag{16}$$

where $\alpha = 0.0001$ is a weighting factor for the control penalty term, $N$ is the total number of joints, and $\Delta t = 0.04$

#### D.1.3  SWIMMER

The agent in the Swimmer environment moves in water with a viscosity of 0.1 , confined within an $xy$-plane. Each joint can have up to three child joints. The root joint state includes height and 2D world velocity. The reward function is the same as that used in 3D Locomotion.

#### D.1.4  GAP CROSSER

This environment presents a unique challenge where the agent must navigate across periodic gaps on an $xz$-plane. The gaps have a width of 0.96 , with a period of 3.2 . The terrain height is 0.5 . Similar to the other environments, each joint can have up to three child joints, and the root joint state includes height, 2D world velocity, and a phase variable encoding the agent's $x$-position. The reward function is defined as:

$$r_t = \frac{|x_{t+1} - x_t|}{\Delta t} + 0.1 \tag{17}$$

with $\Delta t = 0.008$. An alive bonus of 0.1 is also incorporated. The episode terminates when the root height is below 1.0.

### D.1.5 OTHER INFORMATION

Similar to Transform2Act (Yuan et al., 2022), to ensure consistency across different design configurations, each agent is specified using XML strings during the transform stage. The design is represented as an XML string, which is modified based on the transform actions. At the start of the execution stage, the modified XML string is used to reset the MuJoCo simulator and load the newly-designed agent. This approach allows for seamless integration and evaluation of various design modifications within the MuJoCo environment.

## D.2 TETRIS-RELATED TASK

In this part, we provide a comprehensive overview of the two Tetris-related environments used in our experiments.

### D.2.1 TETRIS

In the Tetris environment, the agent manipulates falling blocks to complete horizontal lines without gaps. Each step increments the reward by 1, promoting continuous gameplay, while termination due to a stack reaching the top incurs a penalty of -100. During the design stage, the agent designs four distinct blocks, providing diverse shapes to enhance gameplay. The objective is to optimize these designs to improve performance in Tetris. Mathematically, the reward function is expressed as:

$$r_t = \begin{cases} 1 & \text{if the game continues,} \\ -100 & \text{if the game terminates.} \end{cases} \tag{18}$$

In practice, the maximum steps for each Tetris game round is set to 128, meaning the optimal score for each round is 128. Our method successfully identifies blocks enabling indefinite gameplay in Tetris.

We model the design optimization of Tetris as a multi-step MDP, which can be directly handled by RL methods:

**Design Stage**   In this stage, the agent designs $k = 4$ Tetris blocks, each represented as a $3 \times 3$ grid with 4 squares filled. The state at time $t$ is denoted by $s_t \triangleq (d_t, o_t)$, where $d_t$ is the current design, $t$ is the time step, and $o_t$ is the task/environment state. The action $a_t$ involves selecting and placing the squares in the $3 \times 3$ grid to form a valid Tetris block.

The policy $\pi(a_t \mid s_t)$ maps the state to actions, defined as:

$$\pi(a_t \mid s_t) \triangleq p_\theta(x_{t+1} \mid d_t, o_t) \tag{19}$$

where $p_\theta$ is the probability distribution over the actions conditioned on the current state and design.

The transition probability $P(s_{t+1} \mid s_t, a_t)$ is given by:

$$P(s_{t+1} \mid s_t, a_t) \triangleq (\delta_{d_t}, \delta_{x_{t+1}}, \delta_{o_t}) \tag{20}$$

where $\delta$ denotes the Dirac delta function, ensuring deterministic transitions between states based on the selected actions.

The initial state distribution $\rho_0(s_0)$ is defined as:

$$\rho_0(s_0) \triangleq (p(d_0), p(o_0)) \tag{21}$$

where $p(d_0)$ is the initial design distribution and $p(o_0)$ represents the initial environment state/observation distribution.

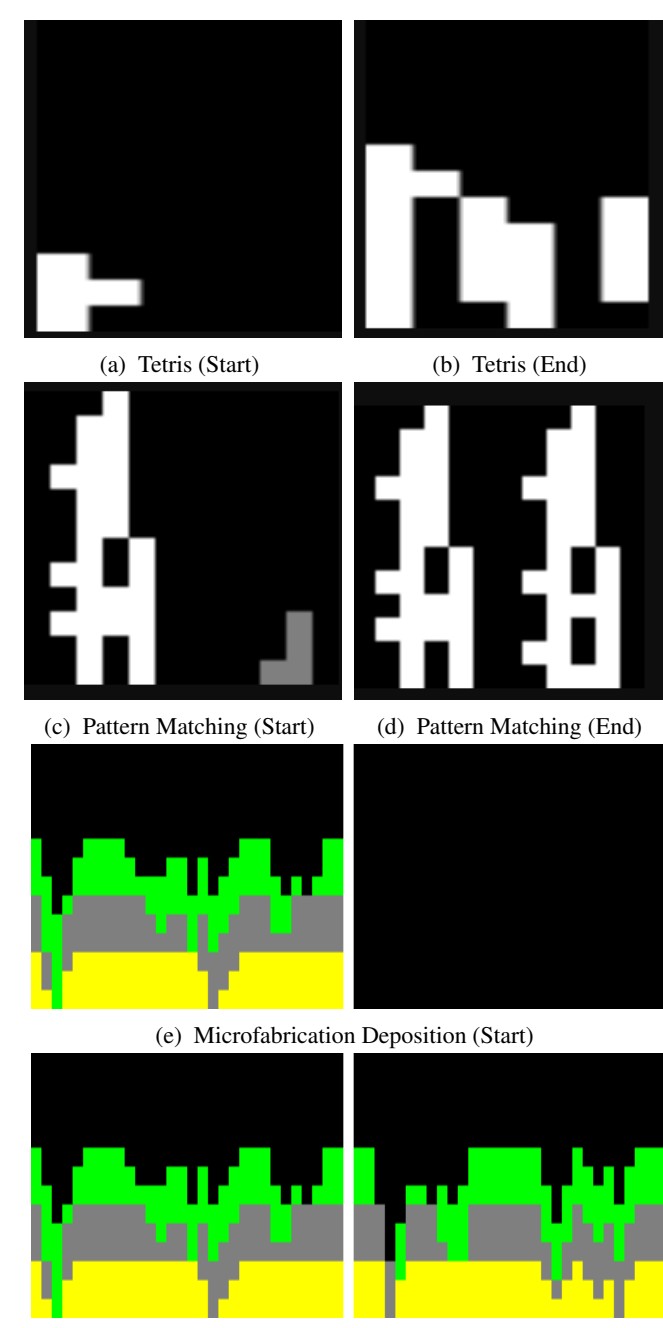

(a) Tetris (Start)      (b) Tetris (End)

(c) Pattern Matching (Start)      (d) Pattern Matching (End)

(e) Microfabrication Deposition (Start)

(f) Microfabrication Deposition (End)

Figure 9: Our agent in each of Tetris-like tasks. In the pattern matching task (i.e., (c) and (d)). The left is the target pattern and the right is the one constructed by the agent using designed blocks. **For the Microfabrication Deposition task, the goal of our agent is similar to the pattern matching tasks where the Target chip structure (i.e., target pattern) is given by human experts. And the design blocks have larger dimension and design space.** Even so, our algorithm can still complete the automated design of the target structure very well, see (e) and (f).

**Control Stage** After designing the Tetris blocks, the agent enters the control stage, where the objective is to play the Tetris game using the designed blocks. The control stage is modeled similarly to the execution stage in a standard MDP framework.

In the control stage, the state $s_t$ includes the current game board configuration and the current Tetris block being placed. The action $a_t$ involves moving and rotating the Tetris block to place it on the board.

The policy $\pi_c(a_t \mid s_t)$ maps the state to control actions, defined as:

$$\pi_c(a_t \mid s_t) \triangleq p_\theta^c(a_t \mid s_t, d_t) \tag{22}$$

where $d_t$ is the design of the Tetris block and $p_\theta^c$ is the probability distribution over the control actions.

The transition probability $P(s_{t+1} \mid s_t, a_t)$ is determined by the game dynamics:

$$P(s_{t+1} \mid s_t, a_t) = T^c(s_{t+1} \mid s_t, a_t) \tag{23}$$

where $T^c$ represents the transition function of the Tetris game.

The initial state distribution $\rho_0^c(s_0)$ is defined by the initial game board configuration and the first Tetris block to be placed.

The reward function $R_c(s_t, a_t)$ in the control stage is given by the game score obtained by clearing lines:

$$R_c(s_t, a_t) \triangleq r_c(s_{t+1}) \tag{24}$$

where $r_c$ is the reward function of the Tetris game.

The overall objective in the control stage is to maximize the cumulative reward, which corresponds to achieving the highest possible score in the Tetris game using the designed blocks.

### D.2.2   PATTERN MATCHING (MICROFABRICATION)

The Pattern Matching environment challenges the agent to arrange blocks to match a target pattern within a grid. The reward is based on the success of the matching process, with a matching rate of 1 for a perfect match. During the design stage, the agent designs four different blocks to achieve various target patterns. The objective is to optimize these designs to improve the agent's ability to accurately and efficiently match the given patterns. The reward function is defined as:

$$r_t = \text{matching\_rate}(s_t, g) \tag{25}$$

where $s_t$ represents the state of the grid at time $t$, and $g$ is the target pattern. The matching rate measures how well the current grid state matches the target pattern, with a maximum value of 1 for a perfect match. In our experiments, our method achieves a matching rate of approximately 97%.

**Design Stage of Pattern Matching**   In the design stage, the agent designs $k = 4$ different pattern blocks. Each block is a $3 \times 3$ grid where the agent places squares to form specific patterns.

The state at time $t$ is represented by $s_t \triangleq (d_t, o_t)$, where $d_t$ denotes the current design, and $o_t$ represents the state of the task/environment. The action $a_t$ at time $t$ involves selecting and placing the squares in the $3 \times 3$ grid to form a valid pattern block.

The policy $\pi(a_t \mid s_t)$ maps the state to actions, defined as:

$$\pi(a_t \mid s_t) \triangleq p_\theta(x_{t+1} \mid d_t, o_t) \tag{26}$$

where $p_\theta$ is the probability distribution over the actions conditioned on the current state and design.

The transition probability $P(s_{t+1} \mid s_t, a_t)$ is given by:

$$P(s_{t+1} \mid s_t, a_t) \triangleq (\delta_{d_t}, \delta_{x_{t+1}}, \delta_{o_t}) \tag{27}$$

where $\delta$ denotes the Dirac delta function, ensuring deterministic transitions between states based on the selected actions.

The initial state distribution $\rho_0(s_0)$ is defined as:

$$\rho_0(s_0) \triangleq (p(d_0), p(o_0)) \tag{28}$$

where $p(d_0)$ is the initial design distribution, and $p(o_0)$ represents the initial environment state/observation distribution.

The reward function $R(s_t, a_t)$ in the design stage is defined as:

$$R(s_t, a_t) \triangleq \begin{cases} r(d_T) = \text{matching\_rate}(d_T, g) & \text{if } t = T, \\ 0 & \text{otherwise} \end{cases} \tag{29}$$

where $r(d_T)$ evaluates the quality of the final design $d_T$.

We model the design optimization of Pattern Matching as a multi-step MDP, which can be directly handled by RL methods:

**Control Stage of Pattern Matching Task**    After designing the pattern blocks, the agent enters the control stage, where the objective is to match the designed patterns with a target pattern. This stage is modeled similarly to the execution stage in a standard MDP framework.

In the control stage, the state $s_t$ includes the current target pattern configuration and the current pattern block being placed. The action $a_t$ involves selecting and placing the designed pattern block onto the target grid.

The policy $\pi_c(a_t \mid s_t)$ maps the state to control actions, defined as:

$$\pi_c(a_t \mid s_t) \triangleq p_\theta^c(a_t \mid s_t, d_t) \tag{30}$$

where $d_t$ is the design of the pattern block, and $p_\theta^c$ is the probability distribution over the control actions.

The transition probability $P(s_{t+1} \mid s_t, a_t)$ is determined by the pattern matching dynamics:

$$P(s_{t+1} \mid s_t, a_t) = T^c(s_{t+1} \mid s_t, a_t) \tag{31}$$

where $T^c$ represents the transition function of the pattern matching task.

The initial state distribution $\rho_0^c(s_0)$ is defined by the initial target pattern configuration and the first pattern block to be placed. The overall goal in the control stage is to maximize the matching rate by optimally placing the designed blocks on the grid.

### D.3    MICROFABRICATION DEPOSITION TASK

The Deposition environment is similar to the Pattern Matching task, but the agent must match three layers of different colours instead and *remove* material as opposed to adding. The agent still only has access to four blocks in the problem. We use a delta reward function for the reward signal:

$$r_t = \text{matching\_rate}(s_t, g) - \text{matching\_rate}(s_{t-1}, g), \tag{32}$$

where again $s_t$ is the grid's state at time $t$ and $g$ is the three-layer pattern of interest. This reward led to better matching rates for this task in practice. Otherwise, both the control phase and the design phases behave the same in the pattern-matching environment.

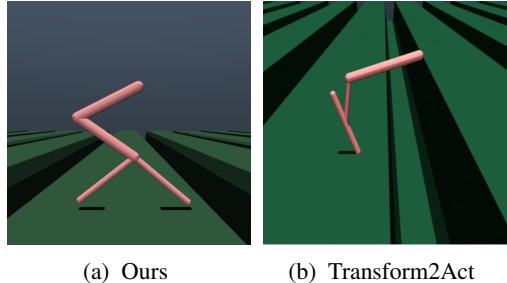

(a) Ours   (b) Transform2Act

Figure 10: Best Design Found in Gap Crosser Task.

# E  BEST DESIGN FOUND BY OUR METHOD

In this section, we would like to share, analyze and interpretate some good design our method found in different tasks.

## E.1  GAP CROSSER

In Figure 10, the two designs for the Gap Crosser task exhibit significant differences in morphology, which impact their performance in navigating the environment's periodic gaps. Our design (See Figure 10a), which features a bipedal form, offers several advantages over the design discovered by Transform2Act (See Figure 10b). Let's analyze these differences and their implications in detail.

**Reach and Stride Length** The elongated limbs in our design significantly enhance the robot's reach, allowing it to span wider gaps with each step. The increased stride length means the robot can cover more ground with fewer steps, which is a critical advantage in a task where efficiency and speed are paramount. The extended reach also reduces the number of transitions the robot needs to make, minimizing the risk of falling.

The Transform2Act design, with its shorter limbs, has a limited stride length. This limitation forces the robot to take more steps to cross the same distance, increasing the number of times it must navigate the gap edges. The shorter reach means that the robot has to exert more effort to span the gaps, which can slow down its progress and increase the likelihood of falling.

**Joint Flexibility and Movement Efficiency** Our design incorporates strategically placed joints that enhance flexibility and movement efficiency. The joints are positioned to allow smooth, natural movements that mimic a walking gait, which is highly efficient for crossing gaps. This flexibility helps the robot adjust its stride dynamically based on the size and distance of the gaps, providing adaptability that is crucial for success in this task.

The Transform2Act design's joint configuration does not optimize movement efficiency to the same extent. The joint angles and placements may restrict fluid motion, making it harder for the robot to adjust its stride effectively. This rigidity can lead to jerky movements and less efficient navigation, reducing the overall performance in the Gap Crosser task.

**Energy Efficiency** The bipedal form of our design promotes energy-efficient movement. The upright posture and long limbs mean the robot can use momentum effectively, reducing the energy required for each step. This efficiency allows the robot to maintain higher speeds and cover more distance without exhausting its energy reserves quickly.

In contrast, the Transform2Act design's lower, more compact form likely requires more energy to lift and move each limb, especially when navigating gaps. The increased energy expenditure can slow down the robot over time, making it less effective in completing the task within a given time frame.

**Adaptability to Terrain** Our design's adaptability to different terrain conditions is another critical advantage. The bipedal structure can easily adjust to varying gap sizes and irregularities in the

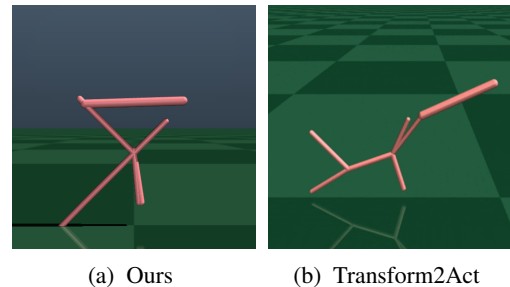

(a) Ours          (b) Transform2Act

Figure 11: Best Design Found in 2D Locomotion Task.

terrain, providing robust performance across different scenarios. This adaptability ensures consistent performance regardless of changes in the environment.

The Transform2Act design may struggle with adaptability due to its less versatile morphology. The limited reach and less flexible joints make it harder for the robot to adjust to unexpected changes in gap size or terrain irregularities, reducing its overall effectiveness in dynamic environments.

In general, our bipedal design offers superior stability, reach, movement efficiency, energy efficiency, and adaptability compared to the design found by Transform2Act. These advantages make our design more suitable for the Gap Crosser task, as it can navigate the gaps more effectively, maintain higher speeds, and adapt to varying terrain conditions. The strategic placement of joints and the elongated limbs contribute significantly to these improvements, showcasing the efficacy of our multi-step MDP approach in optimizing robotic morphology for specific tasks.

### E.2   2D LOCOMOTION

In the 2D Locomotion Task, our design (Figure 11a) outperforms the design discovered by Transform2Act (Figure 11b) due to several key factors. Our design features a more streamlined morphology with one fewer joint on the tail foot and an additional joint on the forelimb, resulting in a more efficient structure for the given task.

Firstly, reducing the number of joints on the tail foot from two to one eliminates unnecessary weight and complexity. This simplification allows the robot to achieve a more stable and balanced gait, crucial for efficient locomotion. The tail foot in our design acts more like a stabilizer, providing necessary support without contributing excess weight that could hinder movement. This contrasts with the design by Transform2Act, which includes an extra hind limb that adds weight and complexity without significant benefits to the locomotion task.

Secondly, the addition of a joint to the forelimb in our design, increasing it from two to three joints, enhances the robot's ability to maneuver and adapt to various terrains. This increased flexibility in the forelimb joints allows for more refined control of movement, improving the robot's ability to propel itself forward efficiently. The added joint provides greater range of motion and better shock absorption, which is particularly beneficial in maintaining high-speed locomotion while minimizing energy expenditure.

Additionally, the overall morphology of our design promotes a more effective distribution of force and balance during movement. The simplified tail structure reduces drag and the potential for destabilizing forces, while the enhanced forelimbs improve traction and propulsion. This combination ensures that the robot can maintain a steady and efficient forward motion, optimizing its velocity and stability. In comparison, the design by Transform2Act suffers from having an additional hind limb that does not significantly contribute to forward propulsion. This extra limb increases the complexity of movement and can lead to inefficient energy usage. Furthermore, the lack of an additional joint in the forelimb limits the range of motion and adaptability of the robot, making it less suited to handle diverse locomotion challenges. In general, our design excels in the 2D Locomotion Task due to its streamlined structure, enhanced forelimb flexibility, and overall balanced morphology. These features collectively contribute to a more efficient and stable movement, allowing the robot to perform the task more effectively than the design discovered by Transform2Act.

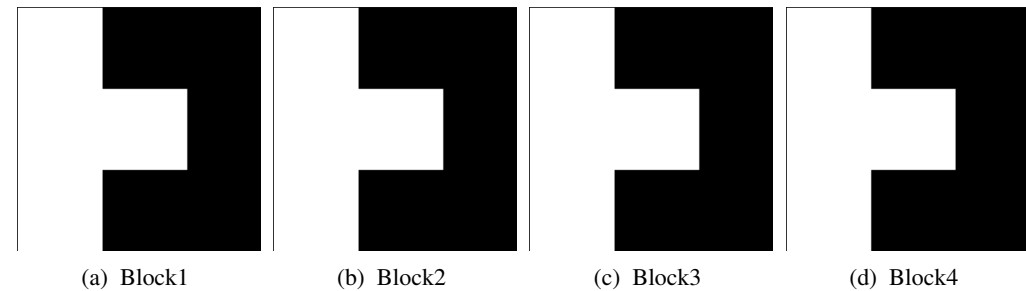

(a) Block1      (b) Block2      (c) Block3      (d) Block4

Figure 12: Best Design Found in Tetris by Our method.

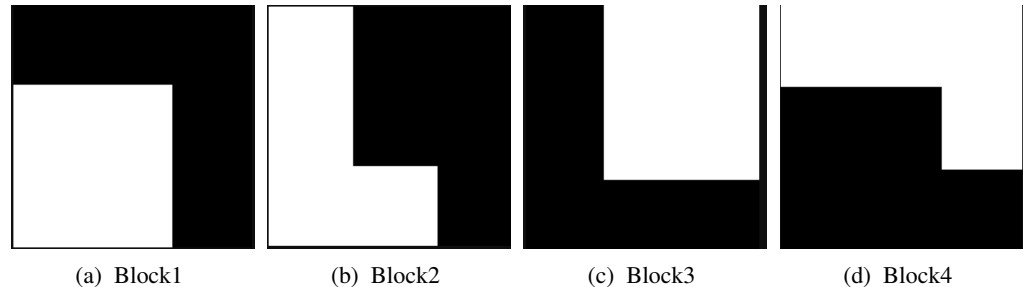

(a) Block1      (b) Block2      (c) Block3      (d) Block4

Figure 13: Best Design Found in Tetris by Transform2Act (Yuan et al., 2022).

### E.3 TETRIS

In the Tetris environment, the agent is tasked with manipulating falling blocks to complete horizontal lines without gaps. The primary goal is to maximize the number of completed lines while avoiding the stack reaching the top of the playing field, which would end the game. The design stage involves creating four distinct blocks, each intended to optimize the agent's performance in achieving this goal.

In the comparison between the optimal designs found by our method (Figure 12) and those found by Transform2Act (Figure 13), several key differences highlight why our designs are superior for the Tetris task.

**Uniformity and Symmetry** Our method produced four identical blocks, each with a symmetrical triangular convex shape. This uniformity is a significant advantage because it simplifies the control strategy for the agent. With identical blocks, the agent can develop a single, effective placement strategy, reducing the complexity of decision-making. In contrast, the designs generated by Transform2Act vary significantly in shape and configuration. This diversity necessitates a more complex control policy, as the agent must account for different shapes and their corresponding placements.

**Efficient Line Completion** The symmetrical triangular convex shape of our blocks allows for seamless interlocking, facilitating the easy formation of complete horizontal lines. This shape minimizes gaps between blocks, which is crucial for preventing the stack from reaching the top of the playing field and terminating the game. The shapes designed by Transform2Act, on the other hand, are less conducive to forming complete lines. The varied and less symmetrical shapes are more likely to create gaps, making it harder to consistently clear lines and maintain continuous gameplay.

**Flexibility and Adaptability** Our uniform blocks provide greater flexibility in placement, accommodating various configurations on the playing field. The symmetrical nature means they can be rotated and placed in multiple orientations, enhancing their utility in maintaining an optimal configuration on the board. This flexibility ensures that the agent can adapt to different scenarios, maintaining continuous gameplay even as the stack of blocks grows. Transform2Act's designs, with their irregular shapes, offer less flexibility and adaptability, making it harder for the agent to handle diverse gameplay situations effectively.

**Continuous Gameplay**    The combination of uniformity, efficient line completion, and flexibility means that our blocks enable the agent to play indefinitely, achieving a perfect score. The optimal control strategy derived from these designs allows the agent to exploit the advantages of the block shapes fully, leading to consistent high performance and maximized rewards. In contrast, the varied shapes from Transform2Act do not support continuous gameplay as effectively. The likelihood of creating gaps and the need for a more complex control strategy reduce the agent's ability to maintain an optimal configuration on the board, leading to more frequent game terminations.

**Simplification of Control Policy**    By using identical blocks, our method reduces the control policy's complexity, as the agent does not need to switch strategies for different shapes. This simplification allows the agent to focus on optimizing the placement of the blocks to maximize line completions, further enhancing performance. Transform2Act's varied block designs require the agent to constantly adapt its control strategy, increasing the likelihood of suboptimal placements and game terminations.

In general, the optimal designs found by our method are superior to those generated by Transform2Act due to their uniformity, symmetry, efficiency in line completion, flexibility, and simplification of the control policy. These attributes collectively enable the agent to maintain continuous gameplay and achieve the highest possible scores in the Tetris task.

# F DETAILED IMPLEMENTATION OF ADAPTIVE CONTROL MECHANISM

A fixed probability $p$ can help agents balance the trade-off between exploration and exploitation. However, it does not allow the agent to adaptively select the most appropriate design method according to different learning stages. For instance, during the early stages of training, agents should actively explore the entire design space by selecting a large exploration rate $p$, rather than spending time exploiting suboptimal designs. Conversely, in the latter stages of training, when sufficient good designs have been discovered and the design space has been thoroughly explored, agents should focus on exploiting these good designs by using a smaller exploration rate $p$.

To address this limitation, we propose a meta-controller that dynamically adjusts the design exploration rate $p$, balancing exploration and exploitation throughout the design optimization process. Specifically, we employ a multi-armed bandit (MAB) approach to help the agent decide whether to design from scratch or use good examples. Each bandit has two arms: arm 0 represents designing from scratch, and arm 1 represents designing from good examples.

In this section, we introduce the adaptive exploration mechanism used in our method, leveraging MAB to dynamically adjust the exploration-exploitation trade-off during the design optimization process.

## F.1 BANDIT-BASED EXPLORATION-EXPLOITATION ADJUSTMENT

Our method leverages a two-armed bandit to dynamically adjust the exploration-exploitation trade-off:

### F.1.1 EXPLORATION-EXPLOITATION CHOICES

In our approach, we simplify the problem by having only two discrete choices for the exploration rate $p$. This results in a two-armed bandit problem, where:

- Arm $k = 0$ corresponds to designing from scratch.
- Arm $k = 1$ corresponds to starting from a good design example sampled from the design buffer.

### F.1.2 SAMPLING AND UPDATING

We employ Thompson Sampling (Garivier & Moulines, 2011) for the MAB implementation. The set of arms $K = \{0, 1\}$ represents the two choices for the design process.

At each round, the actor samples the arm with the highest mean reward. Initially, each actor produces a sample mean from its mean reward model for each arm, selecting the arm with the largest mean. Upon observing the selected arm's reward, the mean reward model is updated.

In general, at each time step $t$, the MAB method chooses an arm $k_t$ from the set of arms $K = \{0, 1\}$ according to a sampling distribution $\mathcal{P}_K$, conditioned on the sequence of previous decisions and returns. The probability distribution for choosing an arm is given by:

$$\mathcal{P}_K = \frac{e^{\text{Score}_k}}{\sum_j e^{\text{Score}_j}} \tag{33}$$

Here, the score for each arm is given by the Upper Confidence Bound (UCB) formula (Garivier & Moulines, 2011):

$$\text{Score}_k = V_k + c \cdot \sqrt{\frac{\log\left(1 + \sum_{j \neq k} N_j\right)}{1 + N_k}} \tag{34}$$

where $V_k$ is the expected value of the returns, and $N_k$ is the number of times $\text{arm } k$ has been selected. This ensures that the agent avoids repeatedly selecting the same arm, thus preventing premature convergence to suboptimal solutions and handling non-stationarity.

**Remark** (Z-Score Normalization). *In practice, Z-score normalization is used to normalize $V_T(x)$ :*

$$\text{Score}_x = \frac{V_T(x) - \mathbb{E}\left[V_T(x)\right]}{D\left[V_T(x)\right]} + c \cdot \sqrt{\frac{\log\left(1 + \sum_j N_T(j)\right)}{1 + N_T(x)}} \tag{35}$$

**Remark** (Design Exploration Rate). *It's worth noting that the design exploration rate, denoted by $p$, is derived from the probability distribution of selecting the 0th arm in our bandit-based approach. This probability distribution is calculated as follows:*

$$p = \mathcal{P}_{(\text{arm}=0)} = \text{softmax}\left(\text{Score}_{arm=0}\right) = \frac{e^{\text{Score}_{k=0}}}{\sum_j e^{\text{Score}_j}} \tag{36}$$

### F.1.3  DYNAMIC ADJUSTMENT

The agent dynamically chooses between exploration and exploitation by sampling an arm at each decision point. This choice adjusts the design strategy based on the accumulated rewards and the frequency of each arm's selection. If the agent selects arm $k = 0$, it designs from scratch. If the agent selects arm $k = 1$, it uses a good example from the design buffer.

## F.2  POPULATION-BASED BANDIT

To address non-stationarity, we employ a population-based MAB approach. We initialize a population $\{B_{h_1}, \ldots, B_{h_N}\}$, where each bandit is indexed by a hyper-parameter $c_i$. The hyper-parameter $c_i$ is uniformly sampled for each bandit.

### F.2.1  POPULATION-BASED SAMPLE

During sampling, each bandit $B_{c_i}$ samples $D$ arms $k_i \in K$ with the top-D UCB scores. We then summarize the selection frequency of each arm and choose the arm $x_j$ selected most frequently. This ensures robust sampling from the most promising regions.

### F.2.2  POPULATION-BASED UPDATE

Using $x_{j,t}$, the agent decides whether to reuse a base design $d_{\text{good}}$ sampled from the design buffer $\mathcal{B}$ or to design from scratch. The agent then applies the design policy $\pi^D$ and the control policy $\pi^C$ to obtain a trajectory $\tau_i$ and the undiscounted episodic return $G_i = \sum_{t=0}^T r_t$. This return $G_i$ is used to update the reward model $V_k$ corresponding to arm $k$.

---

**Algorithm 2** Population-Based Multi-Arm Bandits (Actor-Wise)

---

1: **for** Each Actor $j$ **do**
2:    // Initialize Bandits Population
3:    Initialize each bandit $B_{c_i}$ in the population with different hyper-parameters $c$.
4:    Incorporate each bandit together to form a population of bandits.
5:    **for** each episode $j$ **do**
6:       **for** each $B_{c_i}$ in bandit population **do**
7:          Sample top-D UCB Score arms via equation equation 35.
8:       **end for**
9:       Summarize the selected arms and count the frequency of each arm.
10:      Uniformly sample an arm $x_j$ among the most frequently selected arms.
11:      Decide whether to design from scratch ($x_j = 1$) or use a good example ($x_j = 0$).
12:      Execute the chosen design strategy and obtain the return $G_j$.
13:      **for** each $B_{c_i}$ in Bandit Population **do**
14:         Update $B_{c_i}$.
15:      **end for**
16:    **end for**
17: **end for**

---

# G DETAILED IMPLEMENTATION OF THE DESIGN BUFFER

The Design Buffer is a crucial component of our framework, enhancing the efficiency and effectiveness of the design optimization process. This section provides a detailed description of the Design Buffer algorithm, along with its pseudocode.

## G.1 DESIGN BUFFER IMPLEMENTATION

The Design Buffer is initialized with a predefined capacity $N$ and begins as an empty set. As training progresses, it is populated with high-performing designs. Each design $d_i$ is evaluated based on its performance score $F(d_i)$. Designs that meet or exceed a quality threshold are stored in the buffer to ensure only the most effective designs are retained.

During the design stage, the agent decides whether to generate a new design $d_{new}$ from scratch or to sample an existing design $d_{sampled}$ from the buffer. This decision is governed by the meta-controller, which dynamically adjusts the exploration probability $p$. The buffer is continuously updated: when a new high-quality design is identified, it is added to the buffer. If the buffer is at full capacity, the design with the lowest performance score is replaced by the new design, provided $F(d_{new}) > F(d_{min})$, where $d_{min}$ is the design with the lowest score in the buffer.

The designs stored in the buffer are periodically refined and re-evaluated, allowing the agent to iteratively improve upon successful designs.

## G.2 PSEUDOCODE FOR DESIGN BUFFER ALGORITHM

The following pseudocode outlines the operations of the Design Buffer within our framework:

---
**Algorithm 3** Design Buffer Algorithm
---
**Initialize:** Design Buffer $\mathcal{B}$ with capacity $N$
$\mathcal{B} \leftarrow \emptyset$
**for** each training iteration $i$ **do**
  **if** random() $< p$ **then**
    $d_{new} \leftarrow$ generate_design_from_scratch()
  **else**
    $d_{sampled} \leftarrow$ sample_from_buffer($\mathcal{B}$)
  **end if**
  $F(d_i) \leftarrow$ evaluate_design($d_i$)
  **if** $|\mathcal{B}| < N$ **then**
    $\mathcal{B} \leftarrow \mathcal{B} \cup \{(d_i, F(d_i))\}$
  **else**
    $(d_{min}, F(d_{min})) \leftarrow \arg\min_{(d_j, F(d_j)) \in \mathcal{B}} F(d_j)$
    **if** $F(d_i) > F(d_{min})$ **then**
      $\mathcal{B} \leftarrow (\mathcal{B} \setminus \{(d_{min}, F(d_{min}))\}) \cup \{(d_i, F(d_i))\}$
    **end if**
  **end if**
  $p \leftarrow$ update_exploration_rate(meta_controller)
**end for**
---

Below are the detailed descriptions of the functions used in the pseudocode:

- **generate_design_from_scratch():** This function generates a new design from scratch, represented as $d_{new}$.
- **sample_from_buffer($\mathcal{B}$):** This function samples a design $d_{sampled}$ from the Design Buffer $\mathcal{B}$ using a softmax probability based on their performance scores.
- **evaluate_design($d_i$):** This function evaluates a design $d_i$ and returns its performance score $F(d_i)$.
- **update_exploration_rate(meta_controller):** This function updates the exploration rate $p$ using the meta-controller to balance exploration and exploitation.

Initially, the Design Buffer is empty. The agent either generates a new design $d_{\text{new}}$ or samples an existing design $d_{\text{sampled}}$ from the buffer based on the exploration probability $p$. Each design $d_i$ is evaluated, and its performance score $F(d_i)$ is obtained. If the buffer has not reached its capacity, the new design is added. If the buffer is full, the design with the lowest score $F(d_{\text{min}})$ is replaced by the new design if $F(d_i) > F(d_{\text{min}})$. The exploration rate $p$ is dynamically adjusted using the meta-controller to maintain an effective balance between exploration and exploitation.

This detailed implementation ensures efficient reuse of successful designs while continuing to explore new design possibilities, significantly enhancing the design optimization process.

## H    REPRODUCIBILITY

In this section we conclude the main algorithm Pseudocode of our method in 4 and the code release details for our Reproducibility. In fact our method directly build on Transform2Act (Yuan et al., 2022) with an adaptive design-reuse mechanism.

---

**Algorithm 4** EDiSon

---

**Require:** number of training iterations $N$, simple initial design $d_{null}$, initial design $d_0$, design buffer $\mathcal{B}$, bandit MAB, design policy $\pi^D$, control policy $\pi^C$, length of design stage $T$
1: Initialize design policy $\pi^D$ and control policy $\pi^C$
2: Initialize design buffer $\mathcal{B} \leftarrow (design = d_{null}, value = 0)$
3: Initialize training data replay buffer $\mathcal{M} \leftarrow \emptyset$
4: **for** iteration $i = 1$ to $N$ **do**
5:     **while** not reaching batch size **do**
6:         **for** jth trajectory $\tau_j$ **do**
7:             // Design Stage
8:             Sample arm $k_j$ from the bandit MAB;
9:             **if** $k_j = 0$ **then**
10:                 $d_0 \leftarrow d_{null}$                                                  ▷ Design from scratch;
11:             **else**
12:                 $d_0 \leftarrow$ Sample from Buffer($\mathcal{B}$)                       ▷ Design Reuse
13:             **end if**
14:             **for** iteration $t = 1$ to $T$ **do**
15:                 Sample design actions $a_t^d$ using $\pi^D$
16:                 Update design $d_t$ with sampled actions $a_t^d$
17:             **end for**
18:             // Control Stage
19:             Use $\pi^C$ to rollout control trajectory with design $d_T$, obtain trajectory return $G_j$
20:             Store trajectory $j$ in data replay buffer $\mathcal{M} \leftarrow \tau_j$
21:             Update design buffer $\mathcal{B} \leftarrow (design = d_T, value = G_j)$
22:             Update bandit with $(k_j, G_j)$
23:         **end for**
24:     **end while**
25:     Update $\pi^C$ and $\pi^D$ using PPO with samples from $\mathcal{M}$
26: **end for**
27: **return** Optimal design $d^*$, control policy $\pi^C$, design policy $\pi^D$

---

### H.1    CODE RELEASE

Our implementation is built upon the Transform2Act source code (Yuan et al., 2022), which is available at Transform2Act GitHub. We implement our method on this base code by integrating our multi-armed bandit, design buffer and design re-use ideas. The detailed implementation, including the corresponding hyperparameter settings, is provided in the algorithm section of our paper. Notably, due to the presence of the bandit, extensive hyperparameter tuning is unnecessary. Consequently, reproducing our method using the open-source Transform2Act code is straightforward. We will also publish the relevant code and data upon the paper's officially published.

# I EXPERIMENTAL DETAILS

## I.1 IMPLEMENTATION DETAILS

We employ the Proximal Policy Optimization (PPO) algorithm (Schulman et al., 2017) to learn both the design policy $\pi^D$ and the control policy $\pi^C$. For the robotic morphology design tasks, we use the same network architecture as Transform2Act (Yuan et al., 2022) to ensure a fair comparison. Specifically, we utilize the same Graph Neural Networks (GNNs) to represent both policies, which facilitates generalization across different designs. In the Tetris-related tasks, we adopt a 3-layer Multilayer Perceptron (MLP) to represent all policies and critics.

Our algorithm's code and its detailed pseudocode are provided in App. H. The multi-armed bandit implementation is elaborated in App. F, and the design buffer details are covered in App. G. Comprehensive hyperparameters used in our experiments can be found in App. J.

## I.2 EXPERIMENTAL SETUP

In the robotic morphology design tasks, we follow a setup similar to Transform2Act (Yuan et al., 2022). We capture the undiscounted episode returns averaged over 5 seeds, using a windowed mean across 50,000 environment steps. This setup, along with the default parameters, ensures consistency and comparability of results.

## I.3 RESOURCES USED

All experiments were conducted on a system with one worker equipped with an 8-core CPU and, an NVIDIA V100 GPU, and memory of 32 GB. This setup provided sufficient computational power to train and evaluate our models efficiently. We train our models for three days for the robot morphology design tasks and 4 hours for Tetris-Related Tasks.

## J    HYPERPARAMETERS

In this section, we outline the hyperparameters we used for Efficient Design and Stable Control (EDiSon) and the baseline model, Transform2Act (Yuan et al., 2022). Similar to Transform2Act, our implementation is based on PyTorch and utilizes the PyTorch Geometric package for handling Graph Neural Networks (GNNs). Specifically, we also employ GraphConv layers. To train our policies, we use PPO with Generalized Advantage Estimation (GAE) (Schulman et al., 2017).

### J.1    HYPERPARAMETERS FOR OUR METHOD

For Efficient Design and Stable Control (EDiSon), we conducted a thorough hyperparameter search to ensure optimal performance. We trained our policy using a batch size of 50,000 over 1,000 epochs, resulting in a total of 50 million simulation steps. The detailed hyperparameters are summarized in Table 1.

To ensure a fair comparison, we adopt the same GNN architecture and hyperparameters as Transform2Act, which has been detailed in Table. 2. So we won't go into details about this part of hyperparamters, which has been detailed in Transform2Act (Yuan et al., 2022). We adhered to the same total number of simulation steps. Transform2Act was trained with a population of 20 agents, each using a batch size of 20,000 for 125 generations, also amounting to 50 million simulation steps.

Our rigorous approach to hyperparameter selection and training ensures a level playing field in evaluating the performance of Efficient Design and Stable Control (EDiSon) against Transform2Act. By maintaining consistent training parameters, we provide a robust and reliable comparison, highlighting the strengths and capabilities of our method in various design optimization tasks.

Table 1: Hyper-Parameters for Robotic Morphology Design Experiments.

| Parameter | Value |
| --- | --- |
| GAE $\lambda$ | 0.95 |
| Discount factor $\gamma$ | 0.995 |
| Num. of PPO Iterations Per Batch | 10 |
| Total Training Epochs | 1000 |
| Design Buffer Size | 500 |
| Num. of Bandit | 7 |
| PPO clip $\epsilon$ | 0.2 |
| PPO batch size | 50000 |
| PPO minibatch size | 2048 |
| Num. Bandit | 7 |
| Buffer Size | 500 |
| c of Bandits | Uniform(0,2.0) |

### J.2    HYPERPARAMETERS FOR BASELINE

In this section concluded the hyperparameters used for baseline (Transform2Act) in Table. 2 (Yuan et al., 2022).

Table 2: Hyperparameters used by the baseline method Transform2Act. For Gap Crosser, we also use 0.999 for the discount factor $\gamma$.

| Hyperparameter | Selected |
|---|---|
| Num. of Skeleton Transforms $N_s$ | 5 |
| Num. of Attribute Transforms $N_z$ | 5 |
| Policy GNN Layer Type | GraphConv |
| JSMLP Activation Function | Tanh |
| GNN Size (Skeleton Transform) | (64, 64, 64) |
| JSMLP Size (Skeleton Transform) | (128, 128), |
| GNN Size (Attribute Transform) | (64, 64, 64) |
| JSMLP Size (Attribute Transform) | (128, 128) |
| GNN Size (Execution) | (32, 32, 32), (64, 64, 64) |
| JSMLP Size (Execution) | (128, 128) |
| Diagonal Values of $\Sigma^z$ | 0.01 |
| Diagonal Values of $\Sigma^e$ | 1.0 |
| Policy Learning Rate | 5e-5 |
| Value GNN Layer Type | GraphConv |
| Value Activation Function | Tanh |
| Value GNN Size | (64, 64, 64) |
| Value MLP Size | (128, 128) |
| Value Learning Rate | 3e-4 |
| PPO clip $\epsilon$ | 0.2 |
| PPO Batch Size | 50000 |
| PPO Minibatch Size | 512, 2048 |
| Num. of PPO Iterations Per Batch | 10 |
| Num. of Training Epochs | 1000 |
| Discount factor $\gamma$ | 0.995 |
| GAE $\lambda$ | 0.95 |

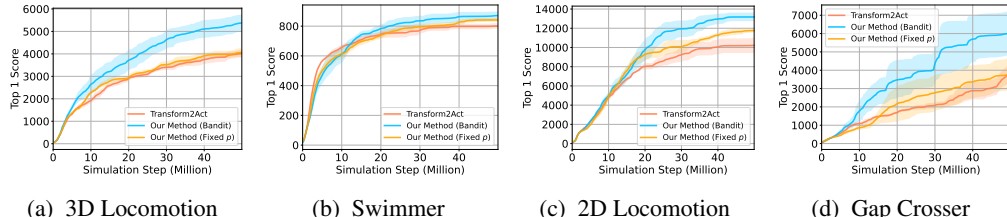

(a) 3D Locomotion      (b) Swimmer      (c) 2D Locomotion      (d) Gap Crosser

Figure 14: **Baseline Comparison (Top-1 Score).** For each robot tasks, we plot the mean and standard deviation of total rewards against the number of simulation steps for all methods. Each curve is smoothed with a moving average over 5 points.

## K    EXPERIMENT RESULTS OF ROBOT-RELATED TASK

### K.1    TOP-1 SCORE

Apart from the average score, we also record the top-k designs scores across the training in Figure 14, where our method with a bandit can find far more better good designs than Transform2Act. For example, In the 3D Locomotion task (Figure 3a), our Bandit method demonstrates a significant advantage over both Transform2Act and our fixed probability $p$ method. The top-1 score for the Bandit approach quickly surpasses that of the other methods, indicating its superior ability to identify and optimize the best designs. The same results show in 2D Locomotion, Gap Crosser and 3D Locomotion in the Water (Swimmer).

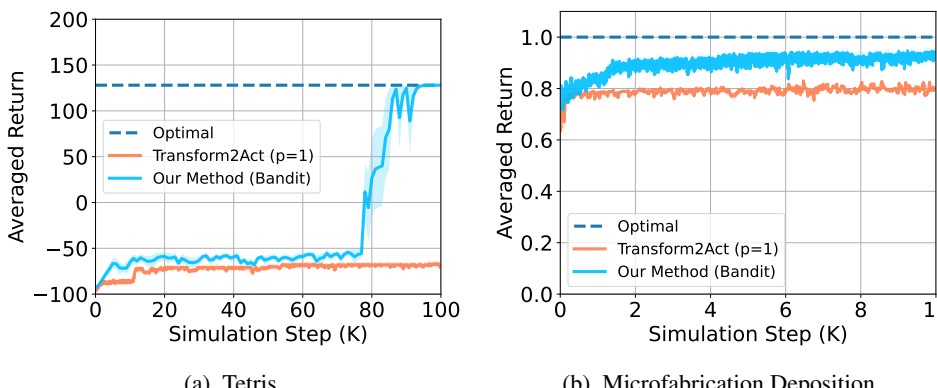

(a) Tetris            (b) Microfabrication Deposition

Figure 15: **Baseline Comparison (Average Return).** For each robot tasks, we plot the mean and standard deviation of total rewards against the number of simulation steps for all methods. Each curve is smoothed with a moving average over 5 points. The pure exploration is a version of Transform2act implementation in Tetris and Microfabrication Deposition Task, i.e., keep others the same as ours and just keep the design exploration rate $p \triangleq 1$, and thus will not reuse learned designs.

## L   EXPERIMENT RESULTS OF TETRIS-RELATED TASK

Our experimental results demonstrate the superior performance of our method compared to the Transform2Act approach across the Tetris and Microfabrication Deposition tasks. These results are illustrated in Figure 15, where the mean and standard deviation of total rewards are plotted against the number of simulation steps for both tasks.

**Tetris**   For the Tetris task (Figure 15a), the curve representing our method shows a rapid increase in average return after approximately 70K simulation steps, eventually reaching a stable and high performance close to the optimal score of 128. This indicates that our method is capable of identifying blocks that enable the agent to play the Tetris game indefinitely, achieving scores that Transform2Act fails to reach. In contrast, Transform2Act maintains a relatively flat curve with modest gains, demonstrating its inability to adapt and optimize as effectively as our approach.

**Microfabrication Deposition**   In the Microfabrication Deposition task (Figure 15b), our method consistently outperforms Transform2Act, as evidenced by the higher average return throughout the entire simulation process. The curve for our method shows a steady increase, approaching the optimal matching rate of 1.0, while Transform2Act plateaus at a lower performance level. This highlights the effectiveness of our bandit-based meta-controller in dynamically balancing exploration and exploitation, which is crucial for achieving high matching accuracy.

The success of our method can be attributed to several key factors. Firstly, our adaptive exploration-exploitation trade-off mechanism allows the agent to efficiently explore new designs and exploit known good designs. This dynamic adjustment is particularly beneficial in complex design tasks, where a static approach like Transform2Act falls short. Secondly, the design buffer in our method facilitates design reuse, enabling the agent to leverage previously successful designs and build upon them. This not only enhances performance but also accelerates the learning process.

Furthermore, our bandit-based meta-controller's ability to adapt to different stages of learning is a significant advantage. Early in the training, the meta-controller promotes exploration to discover a diverse set of designs. As the training progresses and the agent identifies high-quality designs, the meta-controller shifts towards exploitation, refining and optimizing these designs to achieve peak performance.

In general, our experimental results on the Tetris and Microfabrication Deposition tasks showcase the superiority of our method over Transform2Act. The dynamic and adaptive nature of our approach, combined with the efficient design reuse facilitated by the design buffer, leads to significantly better performance and faster learning. These findings underscore the necessity of an adaptive exploration-

exploitation strategy in design optimization tasks and highlight the advantages of our bandit-based meta-controller in achieving superior outcomes.

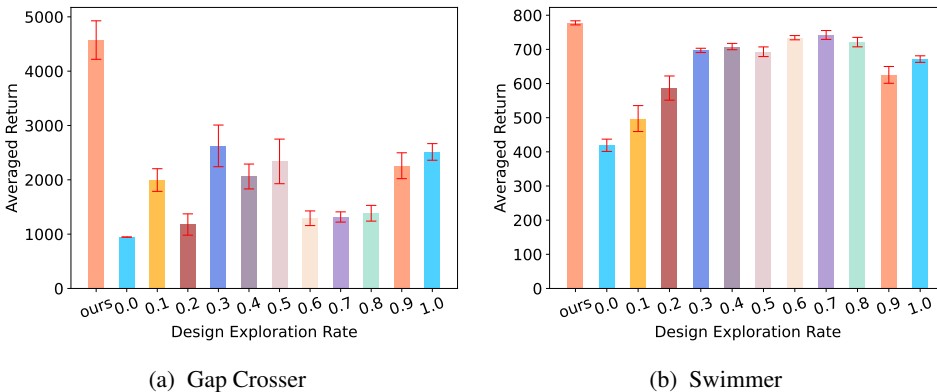

(a) Gap Crosser  (b) Swimmer

Figure 16: **Case Study (Design Exploration Rate Preference).**

## M  CASE STUDY: DESIGN EXPLORATION RATE PREFERENCE

In this section, we present a detailed case study to explore the influence of the design exploration rate on the performance of our proposed method across different tasks. The results, as illustrated in Figure 16, demonstrate that the optimal design exploration rate varies significantly depending on the specific task. This finding underscores the necessity of dynamically adjusting the exploration-exploitation balance to achieve optimal performance.

**Gap Crosser**  For the Gap Crosser task (Figure 16a), the agent shows a clear preference for a design exploration rate around 0.3 to 0.4. At these rates, the agent achieves the highest average return, indicating that a moderate level of exploration allows the agent to discover effective designs while also leveraging previously learned successful strategies. Extremely low or high exploration rates result in suboptimal performance, highlighting the importance of balancing exploration and exploitation. A low exploration rate (e.g., 0.0 to 0.2) limits the agent's ability to discover new and potentially better designs, while a high exploration rate (e.g., 0.8 to 1.0) prevents the agent from fully exploiting known good designs.

**Swimmer**  In the Swimmer task (Figure 16b), the agent's performance peaks at an exploration rate of approximately 0.3 to 0.5. This suggests that, similar to the Gap Crosser task, a moderate exploration rate is most effective. The agent needs to explore sufficiently to find hydrodynamically efficient morphologies while also exploiting designs that have been previously validated as effective. Lower exploration rates fail to provide the diversity of designs necessary for optimal swimming performance, whereas higher rates again hinder the ability to refine and exploit known good designs.

Our findings from these case studies highlight a key advantage of our approach over the Transform2Act method: the ability to dynamically adapt the design exploration rate based on the task at hand. Transform2Act employs a fixed exploration strategy, which may not be optimal for all tasks. The variability in optimal exploration rates across tasks, as evidenced by our experiments, showcases the necessity for an adaptive strategy.

By employing a meta-controller to adjust the exploration rate, our method achieves superior performance across varied tasks. This adaptive strategy allows the agent to explore extensively during the initial phases of learning, ensuring a broad search of the design space, and to shift focus to exploitation in later stages, maximizing the benefits of previously discovered good designs. This balance is crucial in design optimization, where both the discovery of new designs and the refinement of known good designs are necessary for achieving optimal performance.

The case study results clearly demonstrate the task-specific nature of optimal design exploration rates and validate the effectiveness of our adaptive exploration strategy. By allowing the exploration rate to be dynamically adjusted, our method significantly outperforms the fixed strategy employed by Transform2Act (Yuan et al., 2022). This flexibility not only improves the agent's performance in specific tasks but also generalizes well across different types of design optimization problems. The

success of our approach in these diverse tasks underscores the importance of adaptive strategies in reinforcement learning for design optimization, paving the way for more intelligent and efficient design automation in future research.

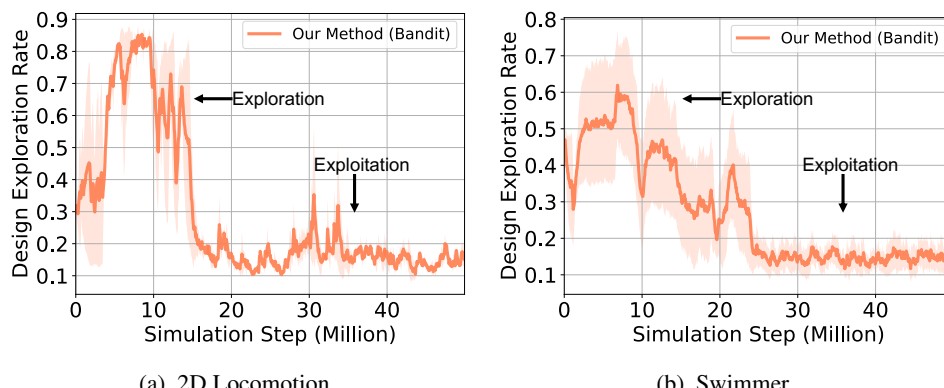

(a) 2D Locomotion  (b) Swimmer

Figure 17: **Case Study (Adatively Exploration-Exploitation Trade-off with Bandit).** For each robot tasks, we plot the mean and standard deviation of design exploration rate against the number of simulation steps for all methods.

## N  CASE STUDY: EXPLORATION-EXPLOITATION TRADE-OFF

In this section, we present a comprehensive case study to demonstrate that our method can adaptively select the appropriate design exploration rate throughout the learning process. The design exploration rate, denoted by $p$, is derived from the probability distribution of selecting the arm=0 in our bandit-based approach. This probability distribution is calculated as follows:

$$p = \mathcal{P}_{(\text{arm=0})} = \text{softmax}\left(\text{Score}_{\text{arm}=0}\right) = \frac{e^{\text{Score}_{k=0}}}{\sum_j e^{\text{Score}_j}} \tag{37}$$

Our case study results, illustrated in Figure 17, demonstrate the effectiveness of our banditbased meta-controller in dynamically balancing the exploration-exploitation trade-off in design optimization problems. The plots show the mean and standard deviation of the design exploration rate across different tasks over the number of simulation steps. This analysis provides insights into how our method adapts to different stages of learning, significantly outperforming the existing Transform2Act method (Yuan et al., 2022) .

**2D Locomotion**    In the 2D Locomotion task (Figure 17a), our method initially emphasizes exploration, with the design exploration rate peaking around 0.7 during the early stages of training. This high exploration rate is crucial for discovering diverse and potentially high-performing designs. As training progresses, the exploration rate gradually decreases, stabilizing around 0.2. This shift signifies a transition towards exploitation, where the algorithm focuses on refining and utilizing the most promising designs discovered during the exploration phase. The adaptive nature of our bandit-based controller allows it to seamlessly navigate between exploration and exploitation, ensuring a balanced approach that maximizes performance.

**Swimmer**    Similarly, in the Swimmer task (Figure 17b), our method starts with a high exploration rate of around 0.6. The exploration rate fluctuates initially, indicating the algorithm's efforts to balance between exploring new designs and exploiting known good designs. As training progresses, the exploration rate stabilizes around 0.2, reflecting a shift towards exploitation. The ability of our method to adjust the exploration rate dynamically is evident in these fluctuations, showcasing its capability to adapt to the changing needs of the task as learning progresses.

**Further Analysis**    The necessity of automatically finding the best design exploration rate for each task is underscored by the variability in optimal exploration rates observed across different tasks. Our bandit-based meta-controller excels in this regard, as it can dynamically adjust the exploration-exploitation balance based on the specific requirements of each task. This adaptability is a significant advantage over fixed-rate methods like Transform2Act, which cannot tailor the exploration rate to the evolving demands of the task.

Compared to Transform2Act, our method demonstrates superior performance in balancing exploration and exploitation. Transform2Act employs a fixed exploration rate, which can lead to suboptimal performance as it cannot adapt to the changing dynamics of the learning process. In contrast, our method leverages a bandit-based meta-controller to dynamically adjust the exploration rate, ensuring that the algorithm can explore extensively during the early stages and exploit effectively in the later stages.

The success of our method can be attributed to its ability to maintain a dynamic balance between exploration and exploitation. By using a meta-controller that adapts the exploration rate based on the observed rewards, our method can efficiently navigate the design space, uncovering high-quality designs and refining them over time. This dynamic adjustment is crucial for optimizing performance across different tasks, as evidenced by the superior results shown in our case study.

Our bandit-based meta-controller effectively manages the exploration-exploitation trade-off, leading to significant improvements in design optimization tasks. The ability to adapt the exploration rate dynamically allows our method to outperform fixed-rate approaches like Transform2Act, highlighting the importance of adaptive strategies in complex design optimization problems.

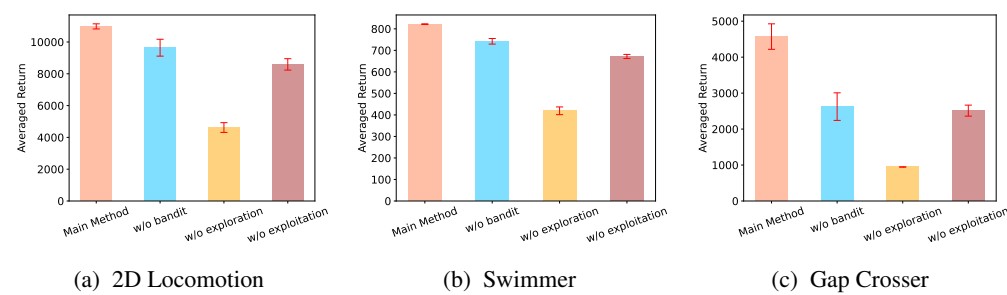

(a) 2D Locomotion       (b) Swimmer       (c) Gap Crosser

Figure 18: **Ablation Study Results (Average Return).**

# O ABALTION STUDIES

In this section, we will provide more details of our abaltion studies.

In our ablation studies, we investigate the importance of two critical components in our approach: the adaptive exploration-exploitation trade-off and the design reuse facilitated by the design buffer. To thoroughly evaluate the impact of these components, we designed several variants of our method:

- **Ours w/o Bandit**: This variant removes the adaptive exploration-exploitation mechanism. The agent is forced to use a fixed exploration rate throughout the training process.
- **Ours w/o Exploitation**: This variant eliminates the design buffer, requiring the agent to always design from scratch. Consequently, it cannot leverage previously successful designs.
- **Ours w/o Exploration**: This variant sets the exploration rate $p$ to 0 throughout the training, effectively disabling exploration and relying solely on exploitation.
- **Our Main Method (with Bandit)**: This is our complete approach, incorporating both the adaptive exploration-exploitation trade-off and the design buffer. The meta-controller dynamically adjusts the exploration rate, balancing between creating designs from scratch and reusing good designs.

The results of these ablation studies are presented in Figure 18. The findings clearly demonstrate the importance of both design reuse and the adaptive exploration-exploitation trade-off. Specifically, the design buffer significantly enhances performance by allowing the agent to leverage previously successful designs, while the adaptive mechanism ensures an effective balance between exploring new designs and exploiting known good ones. Below we will conduct a detailed analysis of the results

**Detailed Analysis** The impact of removing the adaptive exploration-exploitation mechanism (Ours w/o Bandit) was significant across all tasks. This variant showed a notable performance drop, highlighting the necessity of dynamically adjusting the exploration rate. A fixed exploration rate failed to adapt to different stages of learning, leading to suboptimal performance. For instance, in the 2D Locomotion task, the average return was considerably lower compared to our main method, which demonstrates the critical role of the adaptive strategy in efficiently navigating the design space.

Eliminating the design buffer (Ours w/o Exploitation) also resulted in decreased performance. This variant required the agent to design from scratch continuously, preventing it from leveraging previously successful designs. The performance drop observed in tasks such as the Swimmer emphasizes the value of design reuse. Without the ability to reuse effective designs, the agent struggled to maintain high performance, showcasing the necessity of the design buffer in achieving efficient design optimization.

Disabling exploration (Ours w/o Exploration) led to particularly poor performance, especially during the early stages of training. This variant set the exploration rate $p$ to 0, relying solely on exploitation. The results were most evident in the Gap Crosser task, where the average return was significantly lower. The lack of exploration prevented the agent from adequately exploring the design space, limiting its ability to discover high-quality designs. This finding underscores the importance of a balanced approach that includes both exploration and exploitation.

Our main method (with Bandit) consistently outperformed all ablation variants, demonstrating the superiority of integrating both the adaptive exploration-exploitation trade-off and the design buffer. The meta-controller effectively balanced exploration and exploitation, resulting in diverse and high-quality designs across tasks. For example, in the 2D Locomotion task, our main method achieved the highest average return, illustrating its ability to dynamically adjust the exploration rate according to the learning stage. Similarly, in the Swimmer task, the performance was significantly enhanced by the adaptive mechanism, which facilitated the discovery and reuse of optimal designs.

The results of our ablation studies underscore the critical role of adaptive strategies and design reuse in design optimization tasks. The adaptive exploration-exploitation mechanism ensured an effective balance between exploring new designs and exploiting known good ones, while the design buffer allowed the agent to leverage previously successful designs. These components, when combined in our main method, significantly enhanced performance and efficiency. This comprehensive analysis showcases the necessity of an adaptive, task-specific approach to design optimization, further highlighting the superiority of our method over existing approaches such as Transform2Act.

