# OpenReview forum: "EDiSon: Efficient Design-and-Control Optimization with Reinforcement Learning and Adaptive Design Reuse"
_ICLR.cc/2025/Conference — Submitted to ICLR 2025_

### Official Review · Reviewer_gS5Z · 2024-11-01

**Soundness:** 2
**Presentation:** 3
**Contribution:** 2
**Rating:** 3
**Confidence:** 3

**Summary:**

In this paper, the authors study the design-and-control co-design problem. They propose a deep reinforcement learning algorithm to solve this problem. The main innovation is the reuse of previous designs to balance exploration and exploitation in the design space. Experiments on robotic morphology design tasks and a Tetris-based task show improvement over a baseline method.

**Strengths:**

1. The combination of a design buffer and a multi-armed bandit exploration-exploitation tradeoff policy is novel. It also makes intuitive sense why reusing prior good designs can help the optimization progress faster.

2. The paper is mostly well-written and easy to follow.

**Weaknesses:**

1. The experiment section has only a single baseline and the current submission misses several relevant papers [1, 2, 3]. These works all introduce methods for co-design of structure and control policy so their inclusion would strengthen the empirical significance of the submission.

2. While the proposed method is novel, the novelty is limited. The idea is closely related to the experience replay idea which is widely used in deep reinforcement learning algorithms.

[1] Wang, Yuxing, et al. "PreCo: Enhancing Generalization in Co-Design of Modular Soft Robots via Brain-Body Pre-Training." Conference on Robot Learning. PMLR, 2023.

[2] Dong, Heng, et al. "Symmetry-aware robot design with structured subgroups." International Conference on Machine Learning. PMLR, 2023.

[3] Hu, Jiaheng, Julian Whitman, and Howie Choset. "GLSO: grammar-guided latent space optimization for sample-efficient robot design automation." Conference on Robot Learning. PMLR, 2023.

**Questions:**

1. In the paragraph titled “Control As A Multi-Step MDP” (around line 216), is it correct to say that the observation space (both that of the environment and that of the design state) can change based on a specific design? If so, how do the authors ensure that a single control policy is compatible with different observation spaces?

2. Equation 3 seems incorrect. I think the correct formulation is a nested optimization problem
$$d^*=\arg\max_{d} J(\pi_d, d), s.t. \pi_d = \arg\max_{\pi}J(\pi, d)$$

3. In the EDiSon algorithm, the control policy learns from more trajectories as the iteration increases. It is likely that the initial trajectories were poor thus recording lower values for those designs, even if a design is good. This feels like a substantial issue, can the authors comment on this?

---

### Official Review · Reviewer_ehGB · 2024-11-03

**Soundness:** 3
**Presentation:** 3
**Contribution:** 2
**Rating:** 3
**Confidence:** 4

**Summary:**

The paper presents Edison, a new method for automatic design and control. The method is based on the interactive adaptations of the design and controller, with features such as a design buffer to leverage the history of high quality designs encountered during learning.  The method is evaluated on robot morphology and tetris-based design tasks, and is shown to exhibit promising results.

**Strengths:**

The paper tackles an important and relatively under-explored problem of co-designing good  design-controller solutions. The paper is well written, and is fairly simple and easy to understand.

**Weaknesses:**

Some of the design decisions could benefit from better motivation, and should be justified better. Apart from this, the method is compared only with one other baseline. More thorough empirical investigations would be beneficial.

**Questions:**

1.	The results currently only use transform2act as the baseline. However, other relevant methods [1,2] exist, which could be included as baselines or at least discussed in detail.
2.	To expand on the above point, when it comes to co-design, evolutionary methods [3] are a natural choice. Is there a specific reason why the authors have not considered such methods?
3.	In terms of the meta-controller, what motivated the use of an MAB solution? Could other approaches like Bayesian Optimisation have been considered?
4.	Is p in eq 4 fixed? In general, is it not better to anneal it? Since there is mention of using fixed values of p, perhaps it is also worth reporting empirically the effect of different fixed values.
5.	I doubt that setting p=1 is equivalent to transform2act. That approach is fundamentally different, with separate for loops for skeleton, attributes and actions.
6.	In line 279, what do the authors mean by “artificially given good examples”?
7.	As mentioned, a lack of diversity of designs in the design buffer could compromise performance

[1] Luck, Kevin Sebastian, Heni Ben Amor, and Roberto Calandra. "Data-efficient co-adaptation of morphology and behaviour with deep reinforcement learning." Conference on Robot Learning. PMLR, 2020.

[2] Schaff, Charles, et al. "Jointly learning to construct and control agents using deep reinforcement learning." 2019 international conference on robotics and automation (ICRA). IEEE, 2019.

[3] Wang, Tingwu, et al. "Neural Graph Evolution: Towards Efficient Automatic Robot Design." International Conference on Learning Representations.

---

> ### Author Response · Authors · 2024-11-25
> **General response to Reviewre ehGB**
>
> We thank the reviewer for providing insightful comments on our paper.
>
> W1:“Some of the design decisions could benefit from better motivation, and should be justified better. Apart from this, the method is compared only with one other baseline. More thorough empirical investigations would be beneficial.”
>
> We point out that unlike previous work (Transform2Act), we extended our evaluations to additional environments (i.e. Tetris design problem). Previous works have largely focused on robotic locomotion tasks, which we also compare against. We chose to compare against Transform2Act only because it represented the state-of-the-art for joint design-and-control algorithms at time of submission. In their work, their framework surpassed all considered baselines in their analysis and we compared our algorithm on the same set of benchmarks. We include additional comments with the reviewer's questions. We also motivate the addition of the buffer and bandit in our ablation analysis in Figure 7.

---

> ### Author Response · Authors · 2024-11-25
> **Answers to Questions by Reviewer ehGB**
>
> We thank the reviewer for their insightful questions. We included responses below:
>
>
> [Q1] "The results currently only use Transform2Act as the baseline. However, other relevant methods [1,2] exist, which could be included as baselines or at least discussed in detail."
>
> Our conclusion is that the work’s of [1] and [2] although similar in spirit to our work, would not be the most appropriate baselines to compare against because they are restricted to continuous random variables of design. Our method works on mixed distributions of both continuous and categorical variables which makes them incompatible to directly compare against. As is, it is unfair to use them baselines without focused rigorous evaluations on incorporating discrete variables in their frameworks. It is possible that future work might reveal that extending the method’s of  [1,2] to discrete variables is a better alternative approach to joint design and control methods, but that is out of scope to this paper.
>
> [2] To expand on the above point, when it comes to co-design, evolutionary methods [3] are a natural choice. Is there a specific reason why the authors have not considered such methods?
>
> Evolutionary methods are known to be sample inefficient because they do not re-use data collected during design optimization. They also often require larger populations and can be more computationally intensive by evaluating a population of samples in parallel.
>
> In addition, we focused on RL-based approaches for several reasons. First, our method specifically addresses the non-stationary optimization problem created by co-optimizing design and control policies. Second, our framework provides a more principled approach to balancing exploration-exploitation through the bandit-based meta-controller.
>
> [3] In terms of the meta-controller, what motivated the use of an MAB solution? Could other approaches like Bayesian Optimisation have been considered?
>
> We chose MAB for several reasons detailed in Section 5.3. MAB naturally handles the exploration-exploitation trade-off in a non-stationary environment, and our ensemble approach with multiple bandits provides robustness against premature convergence. While Bayesian Optimization could be an alternative, MAB's simplicity and effectiveness in handling non-stationary problems made it particularly suitable for our case.
>
> [4] Is p in eq 4 fixed? In general, is it not better to anneal it? Since there is mention of using fixed values of p, perhaps it is also worth reporting empirically the effect of different fixed values.
> I doubt that setting p=1 is equivalent to transform2act. That approach is fundamentally different, with separate for loops for skeleton, attributes and actions.
>
> No, p is not fixed in our final method. While we present results with fixed p for ablation purposes (Section 6.3), our full method uses an adaptive p controlled by the bandit-based meta-controller. As shown in Figure 6  (line 469) , different tasks have different optimal exploration rates, making adaptive adjustment crucial. The empirical effects of different fixed values are reported in our ablation studies.
>
> The reviewer raises a valid point. While setting p=1 creates similar exploration behavior to Transform2Act, there are indeed structural differences in how designs are generated and modified. We will be more precise in our discussion on the effects of p = 1 in our experiments section for the revision of our paper .
>
>
> [5] “In line 279, what do the authors mean by “artificially given good examples”? As mentioned, a lack of diversity of designs in the design buffer could compromise performance”
>
> By "artificially given good examples," we refer to pre-designed examples that might be provided by human experts or other external sources. Our method instead builds its own repository of good designs through the design buffer (Section 5.2), making it more autonomous and adaptable. We will clarify this terminology.
>
> Furthermore, maintaining a diverse set of designs is an important concern that we address through several mechanisms in Section 5.2. Our design buffer maintains diversity through probabilistic storage based on both performance and diversity metrics. The bandit-based meta-controller further helps prevent premature convergence to a narrow set of designs. Our experimental results demonstrate the effectiveness of these mechanisms in maintaining design diversity.

---

> > ### Comment · Reviewer_ehGB · 2024-11-27
> >
> > Thanks for your responses and clarifications. Just to follow up on that last point - how do you ensure a diverse set of designs? How is this diversity measured? Especially given that "initial diversity" is a crucial factor, it would be good to verify the diversity-related claims via some concrete measure.

---

### Official Review · Reviewer_yQ3c · 2024-11-03

**Soundness:** 3
**Presentation:** 4
**Contribution:** 2
**Rating:** 5
**Confidence:** 4

**Summary:**

This paper addresses the complexities of design optimization tasks, which are often resource-intensive and require specialized expertise. The authors introduce Efficient Design and Stable Control (EDiSON), a reinforcement learning-based approach with minimal human intervention. EDiSON has three key components: 1) a Design Policy that explores the design space step-by-step to efficiently find an optimal design, 2) a Control Policy that optimizes each design for specific tasks, and 3) a Bandit Meta-Controller that balances exploration and exploitation by dynamically choosing between reusing good designs or generating new ones. Experimental results showed that EDiSON outperformed the baseline, Transform2Act, in various design optimization tasks.

**Strengths:**

* The paper is well-structured, and the contributions are clear.
* Using the _adaptive exploration-exploitation balancing_ technique, EDiSON improves sample efficiency, making the approach more practical for real-world applications.
* The ablation study is comprehensive
* EDiSON shows a clear improvement over the baseline (Transform2Act) across various design tasks

**Weaknesses:**

* The novelty of the proposed algorithm feels somewhat limited, as it mainly combines existing methods. The performance improvements therefore seem expected rather than groundbreaking.
* Lack of theoretical analysis on non-ergodic MDP. (But it’s ok, as such papers do not necessarily require theoretical foundations.)
* The single baseline used in the experiment (Transform2Act) makes it challenging to fairly evaluate the efficacy of the proposed method. Could the authors include comparisons with additional recent methods?

**Questions:**

**[Q1]** The control MDP structure described in the paper does not clearly appear to be ergodic. Could the authors elaborate on how EDiSON ensures stability and optimality in non-ergodic environments, either theoretically or experimentally?

**[Q2]** Even though the Bandit-based meta-controller adjusts its exploration dynamically, could there be scenarios where such adaptability might converge prematurely to suboptimal designs? How could one mitigate such risks?

**[Q3]** F(d) (score) can significantly change with each evaluation. Could the authors discuss the potential impact of score variability on the robustness of the design selection process?

---

> ### Author Response · Authors · 2024-11-25
> **Responses to Reviewer yQ3c's comments**
>
> We thank the reviewer for providing insightful thoughts on our paper.
>
>
> W1. “The  novelty of the proposed algorithm feels somewhat limited as it mainly combines existing methods.”
>
> We point out that many major contributions in machine learning have been the result of combining existing techniques in the literature. The most relevant to reinforcement learning is the Atari work of Mnih et al. 2013 [1], which combined replay buffers, convolution networks, and target networks to perform complex tasks. Other highly impactful works include Alex Net [2], which applied convolution networks to image net, and Transformers  [3]. If desired, we can clarify for each how these works combine previously existing methods. Our work, similarly, uses established techniques synthesized together to generate superior performance to prior methods that do not combine these methods.
>
> [1] Mnih, V., Kavukcuoglu, K., Silver, D., Graves, A., Antonoglou, I., Wierstra, D., & Riedmiller, M. (2013). Playing Atari with Deep Reinforcement Learning. arXiv [Cs.LG]. Retrieved from http://arxiv.org/abs/1312.5602
>
> [2] Krizhevsky, A., Sutskever, I., & Hinton, G. E. (2017). ImageNet classification with deep convolutional neural networks. Commun. ACM, 60(6), 84–90. doi:10.1145/3065386
>
> [3] Vaswani, A., Shazeer, N., Parmar, N., Uszkoreit, J., Jones, L., Gomez, A. N., … Polosukhin, I. (2017). Attention Is All You Need. CoRR, abs/1706.03762. Retrieved from http://arxiv.org/abs/1706.03762
>
> W2. “The single baseline used in the experiment (Transform2Act) makes it challenging to fairly evaluate the efficacy of the proposed method. “
>
> At the time of submission, Transform2Act represented the state-of-the-art for joint design generation and control learning algorithms. Across all tasks considered in the previous paper, Transform2Act showed better performance than other baselines. As we evaluated our framework on these same tasks, we concluded it was not beneficial to include additional methods that would only perform worse than Transform2Act and, therefore, would not add additional clarity to the claims in the paper.
>
> We acknowledge that several reviewers have raised this concern as well and refer reviewer yQc3 to our comments to all reviewers above.

---

> ### Author Response · Authors · 2024-11-25
> **Answers to Reviewer yQ3c's Questions**
>
> [Q1] The control MDP structure described in the paper does not clearly appear to be ergodic. Could the authors elaborate on how EDiSON ensures stability and optimality in non-ergodic environments, either theoretically or experimentally?
>
> We agree with the reviewer that given the structure of our problem, the MDP may not be ergodic in practice because it changes as new MDPs are designed via the algorithm. This would mean we may not revisit certain parts of the state space of designs that negatively impact convergence. Despite this, we are not the first to conduct experiments in this context without checking if the MDP is ergodic (see Luck et. al [1] , Yuan et al [2] which are other design and control research papers). We also note that MDPs are generated probabilistically from the transform policy, meaning that there is a non-zero probability to revisit the same design and collect more data.
>
> One of the benefits of EDiSon  is that it makes the design-control optimization problem stable in non-ergodic environments through several mechanisms detailed in Section 5. The design buffer (Section 5.2) maintains a diverse set of high-performing designs, providing stability even when individual designs lead to non-ergodic MDPs. The bandit-based meta-controller (Section 5.3) adaptively balances exploration and exploitation, helping prevent the system from getting stuck in suboptimal regions. Our ensemble approach using multiple bandits with different hyperparameters helps maintain robustness across varying environments. While we acknowledge the lack of theoretical guarantees in non-ergodic settings, our experimental results in Section 6 demonstrate robust performance across diverse non-ergodic environments.
>
> References
> [1] Luck, Kevin Sebastian, Heni Ben Amor, and Roberto Calandra. "Data-efficient co-adaptation of morphology and behaviour with deep reinforcement learning." Conference on Robot Learning. PMLR, 2020.
>
>
> [2] Ye Yuan, Yuda Song, Zhengyi Luo, Wen Sun, & Kris Kitani (2021). Transform2Act: Learning a Transform-and-Control Policy for Efficient Agent Design. CoRR, abs/2110.03659.
>
>
> [Q2] "Even though the Bandit-based meta-controller adjusts its exploration dynamically, could there be scenarios where such adaptability might converge prematurely to suboptimal designs? How could one mitigate such risks?"
>
> It is possible we could converge to sub-optimal designs in practice. In this paper, we use a two-arm UCB bandit for designing from scratch or designing from the stored designs in the design memory. The theoretical guarantee on the regret of the UCB here should be related to the utility distribution of the two arms, which depends on the learning of the design policy in the context of our work. Thanks to UCB, we will always have non-zero probabilities for the arms. However, the optimality depends on the design policy, which is a deep RL policy, where a theoretical guarantee is widely known to be non-trivial to prove for deep RL.
>  Additionally, we use an ensemble of bandits to mitigate issues of overfitting to any single learned bandit. Appropriate selection of different hyperparameters provides robustness against premature convergence. The experimental results in Section 6.3, particularly Figure 6c, demonstrate that our method maintains healthy exploration throughout training while gradually shifting towards exploitation.
>
>
> [Q3] F(d) (score) can significantly change with each evaluation. Could the authors discuss the potential impact of score variability on the robustness of the design selection process?
>
> Issues of score variance are motivations for the inclusion of a design replay buffer and adaptive exploration strategy. A fundamental issue in training a joint design-and-control algorithm is that the estimates of F(d) evolve for evaluated designs. By using a replay buffer, we mitigate this by re-visiting promising designs as a means of re-evaluating their promise as an optimal design.
>
> Furthermore, we handle score variability through several mechanisms detailed in Section 5.2:
> - The design buffer maintains a history of evaluations rather than relying on single scores
> - The probabilistic storage mechanism p(d) ∝ F(d) naturally accounts for score variation
> - The bandit-based meta-controller's UCB scoring helps balance between exploiting consistently high-performing designs and exploring potentially promising but variable designs

---

### Official Review · Reviewer_eZhR · 2024-11-05

**Soundness:** 2
**Presentation:** 2
**Contribution:** 2
**Rating:** 3
**Confidence:** 4

**Summary:**

The paper proposes the use of a design buffer (consisting of previously found good designs) to balance exploration and exploitation in the Transform2Act pipeline. More concretely, it proposes two strategies to utilize this buffer -- the first picks a good design from the buffer with probability $1-p$ and designs from scratch with probability $p$, while the second strategy uses UCB-esque score to decide when to design from scratch. Through experiments on robotic morphology and Tetris-based design tasks, the paper demonstrates the method's efficacy against Transform2Act.

**Strengths:**

- The paper conveys the main ideas clearly.
- The experimental results demonstrate the benefit of utilizing a design buffer.

**Weaknesses:**

- My main concern with this paper is the very incremental nature of the contribution.

**Questions:**

Could the authors elaborate on some key challenges faced when integrating a design buffer into the Transform2Act pipeline, and how their method addresses them? All in all, I am trying to understand if this is straightforward.

---

> ### Author Response · Authors · 2024-11-25
> **Response to Reviewer eZhR's comments**
>
> We thank the reviewer for taking the time to read our paper and provide feedback.
>
> In the context of joint design and control learning, our method is a novel application of both replay buffers and adaptive design sampling. We show in the paper’s ablation analysis that the combination of these mechanisms is key to improving performance. We also note that many important papers could be viewed as straightforward combinations of known methods. For example, the Atari paper was a combination of well-known methods, but their unique combination resulted in important improvements.

---

> ### Author Response · Authors · 2024-11-25
> **Answer's to Questions of review eZhR**
>
> Thank you for your thorough review and for recognizing the clear presentation of our ideas and the demonstrated benefits of our design buffer approach. Let us address your concerns comprehensively:
> Q: "My main concern with this paper is the very incremental nature of the contribution."
> A: While you accurately summarized our method's two strategies for utilizing the design buffer, our contribution goes significantly beyond just adding a buffer to Transform2Act. In Section 4, we introduce a novel theoretical framework that formulates design-and-control as multi-step MDPs, providing fundamental principles for analyzing and improving design optimization methods. This framework addresses the challenging non-stationary optimization problem created by co-optimizing design choice and control policy. The significance of our contribution is demonstrated by dramatic performance improvements - for instance, in the Gap Crosser task, our method achieves a reward of 11,572 compared to Transform2Act's 4,579, representing a 2.5x improvement.
> Q: "Could the authors elaborate on some key challenges faced when integrating a design buffer into the Transform2Act pipeline, and how their method addresses them?"
> A: Integrating the design buffer presented several significant technical challenges:
> First, managing the non-stationary nature of the optimization problem required careful design of both the buffer update mechanism and exploration strategy. As detailed in Section 5.3, we addressed this by implementing an ensemble of MABs with different hyperparameters to handle distribution shifts effectively.
> Second, maintaining an effective balance between design diversity and performance in the buffer was crucial. We solved this through our selective storage mechanism described in Section 5.2, where designs are stored with probability p(d) ∝ F(d).
> Third, the dynamic adjustment of exploration rates across different tasks and training stages presented a significant challenge. Our solution employs a sophisticated bandit-based meta-controller that uses UCB scoring to adaptively balance exploration and exploitation.
> Q: "All in all, I am trying to understand if this is straightforward."
> A: No, implementing these improvements was far from straightforward. While our method can be summarized simply as using a buffer with two strategies (as you accurately noted), the actual implementation required solving complex technical challenges. As detailed in Section 4, we had to develop a comprehensive theoretical framework to properly model the design-and-control problem as multi-step MDPs. This required careful consideration of transition dynamics, reward structures, and policy learning approaches for both design and control phases.
> Our experimental results in Section 6 validate the complexity and effectiveness of our approach through comprehensive ablation studies. Figure 7 shows that removing any single component (bandit mechanism, exploration, or exploitation) significantly degrades performance, indicating that each component addresses a non-trivial aspect of the problem. Furthermore, the case studies in Section 6.3 reveal the intricate relationship between exploration rates and task performance, highlighting why an adaptive approach is necessary.
> The dramatic improvements we achieve across diverse tasks - from robotic morphology design to Tetris-based problems - demonstrate that our method represents a substantial advance in design optimization, providing both theoretical insights and practical improvements that go well beyond incremental changes to Transform2Act.
> We appreciate your thoughtful review and the opportunity to clarify these points. We would be happy to provide additional details about any aspect of our work.

---

### Author Response · Authors · 2024-11-25
**Summary of Reviewer Concerns over Baselines**

We thank all reviewers for providing valuable feedback on our work. We acknowledge that several reviewers have raised concerns over the limited number of baselines considered. We summarize our points we expand in our responses to those specific reviewers:


[1] Baselines were targeted specifically for continuous random variables:  Our method is applicable to domains with mixed distributions of both continuous and discrete random variables. Some suggested baselines only work on continuous random variables, so it is not fair to ad-hoc compare against these methods on discrete random variables which they were not designed for.

[2] Baselines utilize prior information specific to robotics: Some suggested baselines utilise a-priori assumptions which were specific for robotics tasks. Our work is more general as we are interested in design problems beyond strictly robotics. We show this with our results in additional experiments in the Tetris environment. Likewise, since we do not use any priors or inductive biases, these comparisons are not strictly fair where we train both from scratch and without domain-specific knowledge.

[3] Evolutionary algorithms are sample inefficient: As argued in the Transform2Act paper (which also did not compare to evolutionary algorithms), these methods are sample inefficient as they do not reuse data and require immense computation to apply. In high-dimension spaces, they are particularly inefficient.

---

### Meta-Review · Area_Chair_uHGh · 2024-12-21

**Metareview:**

The paper proposes EDiSon, a framework leveraging reinforcement learning for design-and-control optimization, incorporating adaptive design reuse and sequential modeling of design processes. While the approach demonstrates promising results in robotic and Tetris-based tasks, the paper lacks clear differentiation from prior work, limiting its perceived novelty. Additionally, the experimental validation is narrow, failing to demonstrate generalizability across diverse design problems, and the theoretical foundation for the proposed adaptive strategies remains underdeveloped. These weaknesses, particularly the unclear contributions and insufficient validation, lead to the recommendation for rejection.

**Additional Comments On Reviewer Discussion:**

During the rebuttal period, reviewers raised concerns about the paper’s limited novelty, narrow experimental scope, and lack of theoretical analysis for the proposed adaptive strategies. The authors provided clarifications on their contributions and outlined future plans for broader validation but did not sufficiently address the core concerns. The lack of concrete evidence to differentiate the work from prior studies and validate its generalizability weighed heavily in the final decision to recommend rejection.

---

### Decision · Program_Chairs · 2025-01-22

Reject